# IC³: Image Captioning by Committee Consensus

**David M. Chan**[1]*, **Austin Myers**[2], **Sudheendra Vijayanarasimhan**[2],
**David A. Ross**[2], **John Canny**[1,2]
[1]University of California, Berkeley, [2]Google Research
{davidchan,canny}@berkeley.edu, {aom,svnaras,dross}@google.com

## Abstract

If you ask a human to describe an image, they might do so in a thousand different ways. Image captioning models, on the other hand, are traditionally trained to generate a single "best" (most like a reference) caption. Unfortunately, doing so encourages captions that are informationally impoverished. Such captions often focus on only a subset of possible details, while ignoring other potentially useful information in the scene. In this work, we introduce a simple, yet novel, method: "Image Captioning by Committee Consensus" (IC³), designed to generate a single caption that captures details from multiple viewpoints by sampling from the learned semantic space of a base captioning model, and carefully leveraging a large language model to synthesize these samples into a single comprehensive caption. Our evaluations show that humans rate captions produced by IC³ more helpful than those produced by SOTA models more than two-thirds of the time, and IC³ improves the performance of SOTA automated recall systems by up to 84%, outperforming single human-generated reference captions and indicating significant improvements over SOTA approaches for visual description. Code/Resources are available at https://davidmchan.github.io/caption-by-committee.

## 1 Introduction

Generating a high-quality description of an image is not only an open research problem, but it is also a challenging task for humans (Lin et al., 2014; Sharma et al., 2018; Young et al., 2014). Image captioning datasets usually acknowledge this fact; rather than providing a single gold standard caption for each image, they instead rely on human annotators to provide multiple captions for each image, hoping that the set of collected captions collectively captures all of the relevant semantic information.

---
*: Corresponding author

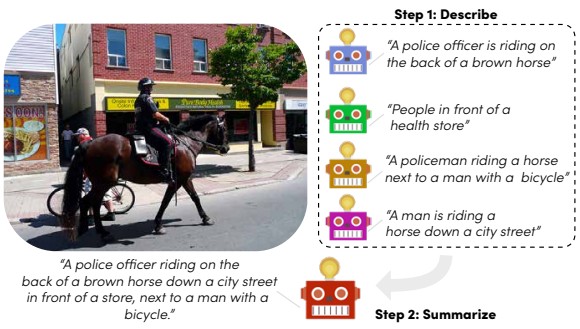

Figure 1: In the IC³ (Image Captioning by Committee Consensus) method, we first leverage standard image captioning models to generate descriptions covering a range of content within the image, similar to how human raters describe the image from independent and unique points of view. We then summarize the group of captions using a vision-free summarization model into a single, high-quality description of the image, suitable for use in visual description applications.

While a set of image captions can be useful, many applications, such as alt-text generation, demand a single succinct sentence that summarizes the information present in the image. This "summarized" caption usually takes a different structural form compared to the "single-viewpoint" captions sourced from crowd workers that make up the datasets. While single-viewpoint captions may contain a subset of the relevant information in an image, it is unlikely that they contain everything (MacLeod et al., 2017; Stangl et al., 2020).

Unfortunately, while the development of large vision and language models (VLMs) has led to progress on a variety of tasks including image captioning, models are trained to produce samples from the reference distribution of a captioning dataset such as MS-COCO (Li et al., 2022; Wang et al., 2022b; Alayrac et al., 2022; Yu et al., 2022; Chen et al., 2022). While not inherently flawed, this approach reproduces the dataset's single annotator viewpoint captions, containing some, but not all, of the semantic information present in the image. Thus we seek to answer the question: "How can we combine many single-viewpoint captions into

a collective summary of the image containing the relevant semantic information?"

One way to obtain a more comprehensive caption, given a set of single-viewpoint captions from annotators, would be to have another human expert consider the set of captions from the committee of individual annotators, and create a new caption that combines complementary information while filtering out any syntactic or semantic errors. Motivated by this idea, we propose the Image Captioning by Committee Consensus ($IC^3$) approach, which utilizes off-the-shelf VLMs in conjunction with large language models (LLMs) to generate higher quality captions than would be possible with VLMs alone. Our key contributions are summarized as follows:

**1.** We introduce $IC^3$, a simple, yet novel, approach leveraging pre-trained image captioning models and large language models to generate semantically complete image captions from a collection of "single-viewpoint" captions.

**2.** We perform an extensive human evaluation of our method which demonstrates that human raters rate image captions generated by $IC^3$ higher in both helpfulness and correctness.

**3.** We demonstrate through several automated measures that captions generated using $IC^3$ contain significantly more semantic information than baseline captions. Notably, CLIP-recall with $IC^3$ captions can be improved by 22-84%, with improved coverage of objects and actions in the image.

## 2 Background and Related Work

The idea that captioning models tend to produce single-viewpoint captions has been prevalent in the image captioning community for many years under several names (Wang and Chan, 2019). Notably, research has focused on quantifying and improving the "diversity" of output captions, including specific methods (Klein et al., 2022; Aneja et al., 2019; Dai et al., 2017; Mahajan et al., 2020; Mahajan and Roth, 2020; Wang et al., 2017, 2016) and metrics (Holtzman et al., 2020; Zhu et al., 2018; Wang et al., 2020; Shetty et al., 2017; Deshpande et al., 2019; Chan et al., 2022b; van Miltenburg et al., 2018; Wang and Chan, 2019). As an alternate approach to increasing and quantifying diversity, some methods (Gan et al., 2017; Yang et al., 2020; Zha et al., 2019; Fang et al., 2022) have focused on explicitly modeling the vari-

ance in the caption space, and introduced human, or statistical controls to reduce the variance, turning the multi-modal problem into several uni-modal problems. While these methods are effective at describing the same image multiple times from multiple perspectives, they have not demonstrated an effective approach that generates a single caption covering all of the information in each of the diverse captions.

Dense captioning methods (Johnson et al., 2016; Yang et al., 2017; Li et al., 2019; Yin et al., 2019; Kim et al., 2019) attempt to generate a full description of all of the objects in the image, however, dense image captions are long and unwieldy, and often contain redundant or repetitive information. Similar long-form captions have been explored in the form of paragraph captioning (Zha et al., 2019; Krause et al., 2017; Liang et al., 2017; Mao et al., 2018; Chatterjee and Schwing, 2018; Luo et al., 2019), however in all cases, no efforts have explored using additional post-processing or models to distill the relevant information for alt-text or for downstream applications. In this work, we explore beyond single-view captioning and move towards captions that are short, succinct summaries of the full visual context.

A natural way of summarizing and filtering a dense caption, paragraph caption, or set of captions, is with a pre-trained model for summarization. While end-to-end methods for abstractive and extractive text summarization exist (Allahyari et al., 2017), recently, large language models (LLMs) such as GPT-3 (Brown et al., 2020), LAMDA (Thoppilan et al., 2022) and PALM (Narang and Chowdhery, 2022) have demonstrated remarkable zero-shot performance when performing language-only summarization tasks (Brown et al., 2020; Liu et al., 2021; Goyal et al., 2022; Chintagunta et al., 2021; Kieuvongngam et al., 2020), so it is natural that they would be capable of summarizing multimodal information in a zero-shot way. Indeed, recently, large-scale vision and language models (VLMs) and large-scale language-only models (LLMs) have revolutionized a number of sub-fields in AI, including in the image captioning space. Models such as BLIP (Li et al., 2022), OFA (Wang et al., 2022b) and Flamingo (Alayrac et al., 2022) have all demonstrated strong performance in single-view image captioning tasks, and indeed, many of these approaches are rated as good or better than human users in some evaluations.

Surprisingly, vision-blind LLMs have also

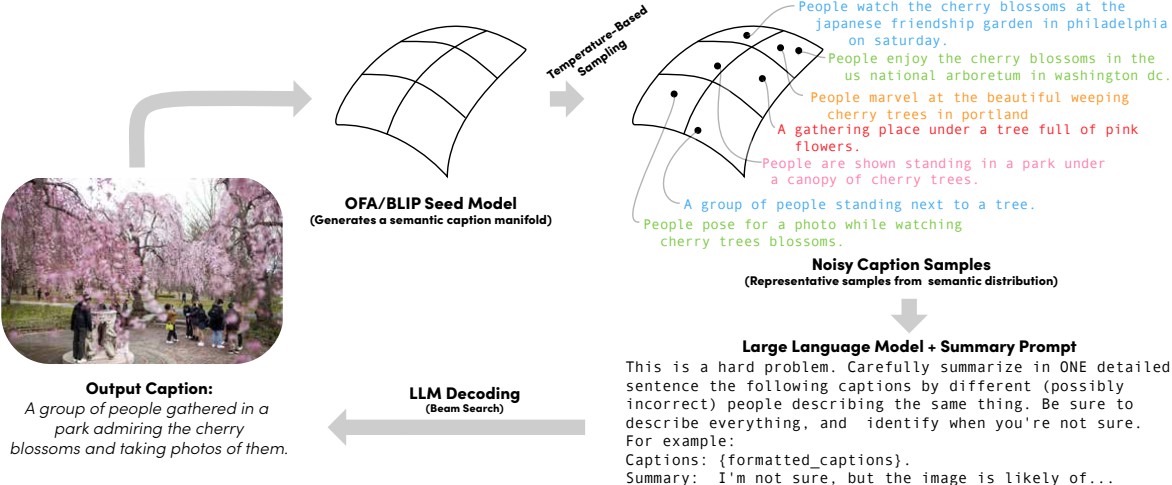

People watch the cherry blossoms at the japanese friendship garden in philadelphia on saturday.

People enjoy the cherry blossoms in the us national arboretum in washington dc.

People marvel at the beautiful weeping cherry trees in portland

A gathering place under a tree full of pink flowers.

People are shown standing in a park under a canopy of cherry trees.

A group of people standing next to a tree.

People pose for a photo while watching cherry trees blossoms.

**Noisy Caption Samples**
(Representative samples from semantic distribution)

**Large Language Model + Summary Prompt**

```
This is a hard problem. Carefully summarize in ONE detailed
sentence the following captions by different (possibly
incorrect) people describing the same thing. Be sure to
describe everything, and  identify when you're not sure.
For example:
Captions: {formatted_captions}.
Summary:  I'm not sure, but the image is likely of...
```

**OFA/BLIP Seed Model**
(Generates a semantic caption manifold)

*Temperature-Based Sampling*

**LLM Decoding**
(Beam Search)

**Output Caption:**
*A group of people gathered in a park admiring the cherry blossoms and taking photos of them.*

Figure 2: The IC[3] approach. Every captioning model defines a distribution across a caption semantic space. This distribution is unlikely to be unimodal, thus, while maximum likelihood decoding approaches such as beam search will capture a local maximum, this point is not likely to be representative of the full distribution of captions. Instead, IC[3] first generates a representative sample of captions from the semantic manifold using temperature-based sampling. This set naturally captures any means as well as the variance of semantic information in the image. Because this group of captions can be large, hard to parse, noisy, or incorrect, we use a large-scale language model, such as GPT-3, paired with prompt engineering, to summarize and filter the noisy group of captions. The resulting captions are more detailed and often more useful than captions generated by beam search alone.

become particularly prevalent in multimodal image/language spaces, primarily using a language-only prefix generated by a set of pre-trained tools. Mokady et al. (2021) explores using a continuous embedding as a prompt for a GPT-style language model and demonstrate strong single-viewpoint image captioning performance, while Hu et al. (2022) and Tiong et al. (2022) leverage natural language prompts along with GPT to achieve SOTA performance on visual question answering.

Closest to our approach are (Zhu et al., 2023) (developed concurrently with the proposed work) and Zeng et al. (2022). Zeng et al. (2022) leverages a CLIP-based model to extract key tags from the image, and then uses GPT-3 along with a specialized prompt to generate a stylized image caption, in an attempt to emulate the Socratic method. Zhu et al. (2023) further employs the Socratic method by employing Chat-GPT and BLIP-2 (Li et al., 2023) to ask and answer questions about the image, respectively. Finally, Chat-GPT summarizes the QA transcript into the image description. Our proposed approach primarily differs from Zhu et al. (2023) and Zeng et al. (2022) in the method of visual data extraction. Instead of using the Socratic method, which requires repeated high-quality questioning and high-quality VQA models to elicit data, or imprecise image tagging models, our approach relies on existing image-captioning models augmented with temperature based sampling, which are able to generate a diverse set of (possibly

noisy) information about the image from multiple sampled viewpoints. This avoids a repetitive (and computationally expensive) QA loop, which with imperfect models can not only introduce significant noise, but also can fail to uncover detail outside the questioning distribution. Also related to our work is Xie et al. (2022), which uses similar tags to generate a paragraph-caption, but does not explore filtering the image, or using existing caption distributions.

## 3  Methods

In this work, we introduce a simple framework for visual description, based on a committee generation then summarization process, which we call "Image Captioning by Committee Consensus" (IC[3]). The approach consists of two stages. In the first stage, we leverage a standard pre-trained image captioning model to sample several (potentially noisy) captions using temperature-based sampling from the caption distribution. This generates a representative samples from the caption distribution, each possibly describing different parts of the image. In the second stage, we leverage a summarization model to summarize the information in each of the captions in a short and succinct way that can be presented to a user. An overview of our method is given in Figure 2.

### 3.1  IC[3]: Image Captioning by Committee Consensus

The goal of IC[3] is to generate an output caption from a given image, by first sampling from a frozen image

captioning model, and then summarizing these generated captions into a single "summary caption". More formally, given an image $I$, we aim to produce a sequence of $m$ tokens $x_1...x_m$ describing the image. Formally, an image captioning model can be described as a function $\mathcal{M}$ which takes $I$ and a set of tokens $a_1...a_{k-1}$ in some vocabulary $V$, and produces a probability distribution $P(a_k \in V|I, a_1...a_{k-1})$, the probability of the next token in the sentence given all previous tokens and the image.

Traditionally, image captioning models generate a caption $C$, where:

$$C = \underset{a_1...a_k \leq N}{\operatorname{argmax}} \prod_{i=1}^{k} P(a_i|a_1...a_{i-1}, I) \qquad (1)$$

Finding the argmax is particularly challenging, as it is an optimization over all possible sequences. Usually, to avoid these challenges, a technique such as beam search (Li et al., 2016; Shen et al., 2017) is used to reduce the number of possible candidates. Recently, however, it has been shown by several papers, including Chan et al. (2022a) and Caglayan et al. (2020) that captions generated using beam search contain only the mutual information between the references, and that such captions are often bland, and uninteresting. To avoid this, we instead take a different approach. Instead of maximizing the likelihood, we generate a set of samples, $K = \{k_1 ... k_i\}$, from the model using temperature-based sampling of the distribution:

$$k_i = a_1...a_{m_i} \propto \exp\left(\frac{\log P(a_1...a_{m_i}|I)}{T}\right) \qquad (2)$$

where $T$ is a temperature parameter. At temperature 1, the resulting samples $K = k_1...k_i$ are an unbiased estimate of the distribution of reference captions. This means that, unlike the maximum likelihood estimate caption, the sampled captions will contain variance in information commensurate with the variance of human descriptions.

Unfortunately, while the caption set $K$ is a good description of all of the details that we might care about in the image, a set of captions can be hard to parse for a downstream user. Thus, we would like to summarize the set $K$ as a single caption, by removing any redundant (or incorrect) information, and combining any details mentioned in one description, and not in one of the others. To do this, we leverage a summarization model $\mathcal{S}$, which maps the set of captions $K$ to our final single output caption $C$. In

IC$^3$, the summarization model is visually blind - that is, the image is not taken into account at all during the summarization phase. We discuss our choice of summarization model in subsection 3.3. In our work, $C$ is generated using beam-search from the summarization model, giving us a maximum likelihood estimate of the best summary of the input captions.

## 3.2 Image Captioning Models

The first stage of the method is to sample a set $K$ of candidate captions from an underlying pre-trained image captioning model, $\mathcal{M}$. In this work, we explore two underlying image captioning engines, the BLIP model (Li et al., 2022), and the OFA model (Wang et al., 2022b), which both represent high-quality image-captioning models pre-trained on large-scale image and language data, then fine-tuned for captioning on the MS-COCO dataset (Lin et al., 2014). More details on the specific image captioning models can be found in Appendix D.1.

**Temperature Selection:** We want to generate a sample of captions that lies as close to the reference distribution as possible. For most models, this will be at or close to temperature 1. To validate this, we use the TRM-CIDEr metric introduced in Chan et al. (2022b) to measure the distance between the reference distribution and the generated captions at a set of temperatures between 0 and 2. We found that for the BLIP model, the optimal temperature was 1.15, and for the OFA model, the optimal temperature was 0.95.

**Selecting size of $K$:** To select the number of captions that are generated, we used a small validation set and found that 10 captions represented a good trade-off between the length of the prompt, and the quality of the model. Sampling larger numbers of candidate captions can improve the total captured information but can decrease the performance of the summarization model, and be more costly to evaluate (See Appendix C.3 for an ablation).

## 3.3 Summarization Models

The choice of the summarization model $\mathcal{S}$ is a key decision when implementing IC$^3$, as the model should be capable of high-quality zero-shot abstractive summarization. We found that using a large language model (LLM) for zero-shot summarization is effective in generating high-quality output summaries of the input captions (See

Appendix C.1). For the results in section 4, we use GPT-3 (Brown et al., 2020), as it produced strong abstractive summaries of the candidate captions, however we discuss (and explore) in detail the choice of summarization model Appendix C.1.

## 3.4 Prompt Selection

To use a large-scale language model for summarization, we follow the techniques in Brown et al. (2020), and design an input text prompt passed to the language model, to encourage the generation of a correct output. The choice of prompt is defined by several motivations, including encouraging the model to summarize the information in the sampled captions, particularly the uncertainty of the captions, encouraging the model to convey this uncertainty in a natural manner (if the model is unsure about what is in the scene, it should identify when this is the case) and making sure that the generated caption is comprehensive, and contains all of the unique viewpoints presented in each of the sampled descriptions. Our final prompt used for experiments with GPT-3 was:

```
This is a hard problem. Carefully
summarize in ONE detailed sentence the
following captions by different (possibly
incorrect) people describing the same
scene. Be sure to describe everything, and
identify when you're not sure. For example:
Captions: {formatted captions}. Summary:
I'm not sure, but the image is likely of...
```

**Encouraging and surfacing uncertainty:** In our prompt design, we aim to encourage the model to account for potential uncertainty/noise in the sampled captions. In many cases, high disagreement among the captions can indicate uncertainty in the language distribution, so we encourage the model to identify when the individual captions differ using language such as `possibly incorrect`, `is likely of` and `I'm not sure` in the prompt. The effect of encouraging and surfacing uncertainty is demonstrated in Table A2 in the appendix. This shows that choosing this language significantly increases the likelihood that models generate uncertain language, and that such captions are rated as more correct on average by human raters.

**This is a hard problem:** Following Kojima et al. (2022), who showed that adding short interrogative/instructive sentences to the beginning of a prompt can improve zero-shot performance, we also add the short sentence "this is a hard problem". We found that this generally improved the quality of the model by a small amount, with diminishing returns as the quality of the candidate captions improved as seen in the ablation in Table A3.

**Use of capitalization:** In our exploration of the prompt space, we found that in some cases, the models choose to generate long concatenations of the input captions instead of generating a single short and concise sentence. Thus, to help alleviate this issue, we found that capitalizing the "ONE" when asking for sentences encouraged the GPT models to produce shortened captions, usually consisting of a single sentence (reducing the average caption length from 120.03 to 107.89 characters).

**Style Transfer and Contextual Captions:** In addition to the above goals, it is interesting future work to explore how the prompt can be used to shape the generated output caption, either for zero-shot transfer to other languages, or to guide the generation of the text to pay attention to specific details in the image. While we do a cursory exploration of such an approach in Appendix H and Appendix I, future work in this area is essential.

## 3.5 Evaluation

n-gram matching scores such as CIDEr (Vedantam et al., 2015) do a poor job at comparing distributions of texts. An example of this is a single caption which is the concatenation of two non-overlapping references. Because for each reference, there exist n-grams in the candidate that do not overlap with that reference, the candidate will score poorly. However the candidate has the same (or more) information than either of the two original reference sentences alone. Thus, along with extensive human evaluation, we introduce two novel automated measures of caption quality, which directly address information retrieval.

**CLIP Recall:** One measure of the quality of a caption is its ability to distinguish between images in a dataset (Hessel et al., 2021; Wang et al., 2022a). In this work, we leverage CLIP (Radford et al., 2021) as a recall model, and use it to estimate an approximate specificity of each of the captions generated by our approach. Specifically, for each image $i$, we compute the CLIP embeddings $\mathcal{I}_i$ and the corresponding caption $\mathcal{C}_i$. We then compute

the CLIP-Score ([Hessel et al., 2021](#)) between $\mathcal{I}_i$, and every other generated caption $\mathcal{C}_j$, and from this, compute the mean reciprocal rank (MRR) of the caption $\mathcal{C}_i$, and the recall at 1, 5 and 10. High values of MRR suggest that the captions are more discriminative within the test set (and thus, are often more detailed in nature).

**Content Coverage:** In addition to the specificity of the individual caption, we also would like to measure how much of the total information provided by *all* of the references is included in the summarized caption. To do this, we first compute the caption $\mathcal{C}_i$ for each image and fetch the references $\mathcal{R}_i^j, 1 \leq j \leq N$ for each image. Let $\mathcal{N}(\mathcal{C}_i)$ be the set of nouns in a caption, $\mathcal{C}_i$, and $\mathcal{V}(\mathcal{C}_i)$ be the set of verbs. Let

$$I_{\mathcal{N},i}(n) = \begin{cases} 1, & \text{if } n \in \cup_{j=1}^N \mathcal{N}(R_i^j) \\ 0, & \text{otherwise} \end{cases} \quad (3)$$

We compute exact noun overlap for $\mathcal{C}_i$ as:

$$\text{Noun Overlap} = \frac{1}{|\cup_{j=1}^N \mathcal{N}(R_i^j)|} \sum_{n \in \mathcal{N}(\mathcal{C})} I_{\mathcal{N},i}(n) \quad (4)$$

Verb overlap is defined analogously for $\mathcal{V}$. We compute fuzzy overlap similar to exact overlap, however instead of [Equation 3](#), we use:

$$I_{\mathcal{N},i}(n) = \begin{cases} 1, & \text{if } ||E(n)-E(x)||_2^2 \leq \phi, x \in \cup_{j=1}^N \mathcal{N}(R_i^j) \\ 0, & \text{otherwise} \end{cases}$$

$$(5)$$

where $E$ is a word-embedding function (we use embeddings from the Spacy package ([Honnibal et al., 2020](#))), and $\phi = 0.1$ is a threshold.

**Human Evaluation:** To test the performance of our model in real-world conditions, we leverage human evaluations on the Amazon Mechanical Turk platform. We perform two styles of human evaluation. In "context-free" evaluation, raters are asked to rate two components: The "Helpfulness" of the caption to somebody who cannot see the image (on a scale of 0 to 4), and the factual "Correctness" of the caption (on a scale of 0 to 5). In "head-to-head" evaluation, raters are presented with two captions and asked which is more "helpful" for somebody who cannot see the image, as well as which is more factually "correct". Full details on the exact questions asked and the experimental design are given in [Appendix F](#).

Table 1: Head-To-Head human evaluation performance of models augmented with $\text{IC}^3$ on the MS-COCO dataset. Table shows % of instances preferred by users.

| Model | Helpfulness ↑ | Correctness ↑ |
|---|---|---|
| Blip-2 + $\text{IC}^3$ | **51.97%** | **44.10%** |
| Blip-2 | 37.49% | 42.67% |
| Tie | 9.78% | 11.6% |
| Blip + $\text{IC}^3$ | **52.05%** | **42.90%** |
| Blip | 33.44% | 36.28% |
| Tie | 14.51% | 20.82% |
| OFA + $\text{IC}^3$ | **52.91%** | **48.93%** |
| OFA | 32.72% | 33.94% |
| Tie | 14.37% | 17.12% |
| Ref + $\text{IC}^3$ | **55.79%** | **48.80%** |
| Ref | 36.65% | 36.27% |
| Tie | 7.46% | 13.97% |

**Reference Baseline:** Because $\text{IC}^3$ can be used to augment any existing captioning method, we also explore augmenting the human reference captions with $\text{IC}^3$. To do this, we use the reference captions (Ref) as candidates for the summary pipeline, which are then summarized by the LLM to generate the Ref+$\text{IC}^3$ caption. Such an approach removes the additional variance introduced by the candidate captioning model and demonstrates the potential of $\text{IC}^3$ applied to a near-perfect captioning approach.

## 4 Results & Discussion

In this section, we compare captions generated using our baseline seed image captioning models, BLIP ([Li et al., 2022](#)), BLIP-2 ([Li et al., 2023](#)), and OFA ([Wang et al., 2022b](#)), to captions generated using $\text{IC}^3$. We leverage two image captioning datasets for evaluation: MS-COCO ([Lin et al., 2014](#)) and the Flickr-30K dataset ([Young et al., 2014](#)) (see [Appendix D.2](#)).

[Figure 3](#) and [Appendix G](#) give some qualitative examples of our method compared to several baseline methods. We can see that descriptions using $\text{IC}^3$ are often longer, and contain more detail than their counterpart baseline models. Further, most display uncertainty about the content of the image in a natural way, which the baselines are not able to accomplish (see [Appendix C.2](#)).

### 4.1 Human Evaluation

Recent works ([Chan et al., 2022b](#); [Caglayan et al., 2020](#)) have confirmed that human evaluation remains the gold standard for visual description evaluation, despite progress in automated evaluation

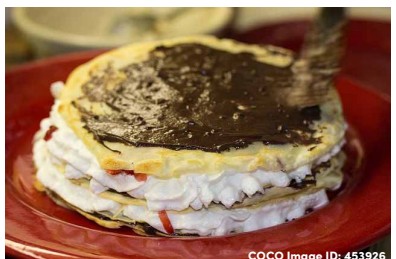 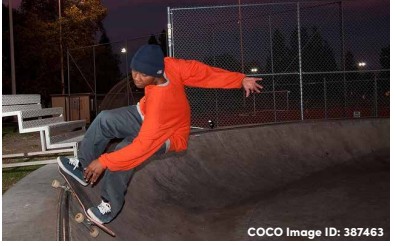 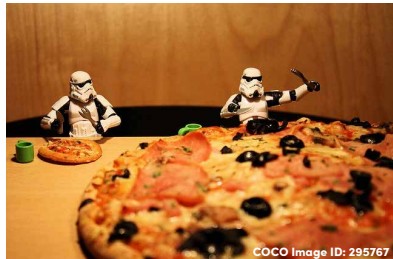

**OFA + IC³:** *A layered cake with chocolate and cream on top, sitting on a red plate, possibly with a knife sticking out of it, although it could also be an ice cream cake, a pancake, a crepe cake, or a stack of pancakes.*
**OFA:** *A cake sitting on top of a red plate.*
**Socratic Models:** *A variety of sweet treats on display at a deli.*
**Human:** *A plate that has a dessert on it.*

**OFA + IC³:** *A young man wearing an orange jacket and hat performing tricks on a skateboard at a skate park or enclosed pool at night.*
**OFA:** *A man riding a skateboard up the side of a ramp.*
**Socratic Models:** *A peaceful scene of a skate park in the city.*
**Human:** *A person doing a skateboard trick up a bowl*

**OFA + IC³:** *Two small figurines, possibly Star Wars related, sitting at table in getting ready to eat a pizza.*
**OFA:** *A couple of figurines sitting on top of a table next to a pizza.*
**Socratic Models:** *A group of people enjoying a delicious pizza at a local pizzeria.*
**Human:** *Two toys are sitting with a toy cup and real pizza.*

Figure 3: Some qualitative examples of IC³ paired with the OFA model. We can see in all of these cases that OFA + IC³ surfaces significantly more detail about the image, including both foreground and background details, as well as maintains the syntactic quality, relevant world-knowledge, and high-level details of the maximum-likelihood caption.

Table 2: Human rater mean opinion score for IC³ on MS-COCO. Helpfulness (H, 0-4), Correctness (C, 0-5).

| CANDIDATE GENERATOR | BASELINE | | + IC³ | |
|---|---|---|---|---|
| | H ↑ | C ↑ | H ↑ | C ↑ |
| OFA | 2.876 | 3.891 | **2.965** | **4.010** |
| BLIP | 2.901 | **3.951** | **2.921** | 3.881 |
| REFERENCES | 2.932 | 3.966 | **2.985** | **3.985** |

Table 3: Head-To-Head human evaluation performance of IC³ on the hard MRR MS-COCO splits.

| MODEL | HELPFULNESS ↑ | CORRECTNESS ↑ |
|---|---|---|
| BLIP + IC³ | **48.06%** | **41.78%** |
| BLIP | 28.50% | 30.43% |
| TIE | 21.01% | 25.60% |
| OFA + IC³ | **51.10%** | **48.90%** |
| OFA | 32.04% | 29.52% |
| TIE | 15.06% | 19.78% |

Table 4: Human rater mean opinion score for IC³ on Hard-MRR subsets. Helpfulness (H, 0-4), Correctness (C, 0-5).

| CANDIDATE GENERATOR | BASELINE | | + IC³ | |
|---|---|---|---|---|
| | H ↑ | C ↑ | H ↑ | C ↑ |
| OFA HARD SUBSET | | | | |
| OFA | 2.452 | 3.651 | **2.713** | **3.713** |
| REFERENCES | 2.649 | 3.675 | **2.728** | **3.902** |
| BLIP HARD SUBSET | | | | |
| BLIP | **2.708** | **3.827** | 2.648 | 3.704 |
| REFERENCES | 2.887 | 3.887 | **2.934** | **3.918** |

of image captioning. As discussed in subsection 3.5, we perform two experiments: head-to-head experiments and mean opinion score evaluation. The results of the head-to-head experiments on MS-COCO are shown in Table 1, where we can see that IC³ augmented models significantly outperform the baselines on both helpfulness and correctness (Helpfulness: OFA + IC³ vs. OFA, $p = 0.0008$; BLIP + IC³ vs. BLIP, $p = 0.008$; BLIP2 + IC³ vs. BLIP2, $p = 0.003$; REF + IC³ vs. REF, $p = 1.73e^{-5}$. Correctness: OFA + IC³ vs. OFA $p = 0.0428$; BLIP + IC³ vs. BLIP, $p = 0.0280$; BLIP2 + IC³ vs. BLIP2, $p = 0.898$; REF + IC³ vs. REF, $p = 0.0019$; $n = 89$).

Table 2 shows the performance of IC³ in terms of mean opinion score, and demonstrates that even in a calibration-free setup, where no extra evidence is presented, IC³ methods significantly outperform their baseline counterparts when rated for helpfulness (Helpfulness: OFA + IC³, $p = 0.0237$; BLIP + IC³, $p = 0.0419$; REF + IC³, $p = 0.0293$; $n = 121$). Numerically, IC³ outperforms baselines on the correctness measure, however we found in all three cases that the difference was not statistically significant. We believe the the reduction in margin is caused by several effects: (1) without a point of reference for the potential quality of the captions, AMT workers cannot tell which captions are deserving of high scores and (2) both OFA and

BLIP are strong captioning models, so a random sample of MS-COCO images may not contain difficult images that separate the two methods.

To investigate this hypothesis, we ran several additional human studies on a set of challenging examples, which we call the Hard MRR splits (see Appendix D.2), which contain the 200 most challenging images for CLIP to recall. We show the head-to-head experiments in Table 3, and see that once again, in head-to-head experiments, IC³ significantly outperforms baseline methods (OFA + IC³, $p = 0.0225, n = 28$, BLIP + IC³, $p = 0.0074, n = 52$). In MOS experiments (Table 4), IC³ augmented OFA and Reference captions both significantly outperform their baselines ($p < 0.05, n = 41$) on both BLIP

Table 5: CLIP Recall for $IC^3$ augmented captions in the MS-COCO Dataset (Karpathy Test Split). MRR: Mean Reciprocal Recall, R@K: Recall @ K.

| MODEL | MRR ↑ | R@1 ↑ | R@5 ↑ | R@10 ↑ |
|---|---|---|---|---|
| REF + $IC^3$ | **0.776** | **0.691** | **0.883** | **0.930** |
| REF | 0.593 | 0.480 | 0.724 | 0.808 |
| OFA + $IC^3$ | **0.748** | **0.656** | 0.857 | 0.914 |
| BLIP + $IC^3$ | 0.734 | 0.639 | 0.848 | 0.908 |
| BLIP2 + $IC^3$ | 0.746 | 0.652 | **0.863** | **0.921** |
| OFA | 0.586 | 0.472 | 0.717 | 0.798 |
| BLIP | 0.501 | 0.382 | 0.634 | 0.736 |
| BLIP2 | 0.589 | 0.473 | 0.725 | 0.811 |

Table 6: CLIP Recall for $IC^3$ captions in the Flickr-30K test set. MRR: Mean Reciprocal Recall, R@K: Recall @ K.

| MODEL | MRR ↑ | R@1 ↑ | R@5 ↑ | R@10 ↑ |
|---|---|---|---|---|
| REF + $IC^3$ | **0.856** | **0.836** | **0.836** | **0.938** |
| REF | 0.708 | 0.679 | 0.679 | 0.798 |
| OFA + $IC^3$ | **0.806** | **0.782** | **0.782** | **0.889** |
| BLIP + $IC^3$ | 0.736 | 0.707 | 0.707 | 0.829 |
| OFA | 0.658 | 0.629 | 0.629 | 0.745 |
| BLIP | 0.499 | 0.463 | 0.463 | 0.581 |

Table 7: Content coverage performance on $IC^3$ augmented captions in the MS-COCO Dataset (Karpathy Test Split). N: Noun Recall, V: Verb Recall

| MODEL | EXACT | | FUZZY | |
|---|---|---|---|---|
| | N ↑ | V ↑ | N ↑ | V ↑ |
| REF + $IC^3$ | **0.552** | **0.354** | **0.767** | **0.616** |
| REF | 0.255 | 0.137 | 0.567 | 0.398 |
| BLIP2 + $IC^3$ | **0.364** | **0.229** | **0.667** | 0.529 |
| BLIP + $IC^3$ | 0.353 | 0.223 | 0.663 | **0.534** |
| OFA + $IC^3$ | 0.351 | 0.211 | 0.656 | 0.498 |
| BLIP2 | 0.277 | 0.185 | 0.582 | 0.442 |
| BLIP | 0.266 | 0.196 | 0.573 | 0.486 |
| OFA | 0.275 | 0.171 | 0.583 | 0.412 |

Table 8: Content coverage performance on $IC^3$ augmented captions in the Flickr-30K Test Dataset.

| MODEL | EXACT | | FUZZY | |
|---|---|---|---|---|
| | NOUN ↑ | VERB ↑ | NOUN ↑ | VERB ↑ |
| REF + $IC^3$ | **0.548** | **0.350** | **0.763** | **0.684** |
| REF | 0.246 | 0.147 | 0.543 | 0.490 |
| BLIP + $IC^3$ | 0.283 | **0.200** | 0.604 | **0.585** |
| OFA + $IC^3$ | **0.296** | 0.195 | **0.607** | 0.571 |
| BLIP | 0.205 | 0.134 | 0.505 | 0.507 |
| OFA | 0.230 | 0.147 | 0.533 | 0.495 |

and OFA Hard MRR sets, but the experiments with BLIP on the BLIP Hard MRR set are inconclusive ($p = 0.682, n = 41$), suggesting that in some cases, $IC^3$ is unable to overcome all of the challenges with the seed captioning model. The fact that the head-to-head performance on the BLIP Hard MRR split in Table 3 is stronger for $IC^3$, coupled with the fact that reference captions augmented with $IC^3$ perform better on this set suggests that $IC^3$ can manage some of the underlying noise, does not fully compensate for a lack of calibration.

## 4.2 Automated Evaluation

As discussed in subsection 3.5, we also perform automated evaluations of the method on both the MS-COCO and Flickr-30K datasets. The performance of $IC^3$ in CLIP recall is first demonstrated in Table 5, where for MS-COCO, CLIP recall MRR is improved by 27.6% under OFA, and by 46.5% under BLIP, suggesting that $IC^3$ augmented captions significantly outperform SOTA captions in indexing scenarios. Similar improvements exist in Table 6, where $IC^3$ improves CLIP MRR by 22.49% for OFA and up to 84.46% for BLIP. These results suggest that $IC^3$ surfaces significant additional detail compared to individual baseline and reference sentences, leading to strong recall performance, and suggesting that $IC^3$ augmented captions can lead to benefits when applied to indexing

and search. **On all datasets, $IC^3$ outperforms single human reference captions**, suggesting that summarizing multiple viewpoints is essential for strong automated recall performance.

Table 7 and Table 8 both demonstrate the summarization ability of $IC^3$ augmented methods, as $IC^3$ outperforms all baseline methods in recalling content from the dataset, often by relatively large margins. The verb recall is often lower (though still improved) across all approaches, suggesting that $IC^3$ focuses recalling content over action in an image. We further quantify $IC^3$'s summarization capability in Appendix C.5 where we find that increasing the diversity of the input candidates can improve noun/verb recall, however has little impact on MRR. These results suggest that $IC^3$ summarizes any salient information as required.

While Appendix D.3 discusses the performance of our methods on N-Gram measures, such measures are relatively misleading, as we generate captions that *differ significantly* from reference captions, thus, the N-Gram metrics are naturally lower compared to maximum-likelihood baselines.

## 5  Limitations

While IC³ significantly outperforms baseline captioning approaches, as well can outperform single human image captioning references, it also suffers from several distinct limitations.

**Hallucination:** While IC³ often produces high-quality summaries of the associated captions, it has several distinct failure modes, mostly coming down to hallucinations induced by the underlying captioning model. In some cases, objects that are hallucinated by the model can propagate to the summary, even if they are internally inconsistent with other captions in the candidate set $K$. Another distinct failure mode of the captions is when uncertainty in the samples is interpreted as two distinct parts of the image. For example, if 50% of the captions refer to a dog, and 50% of the captions refer to a cat, the model may infer that both a dog and a cat are present in the image, even though only a single, unknown, animal is there. Examples of these failure cases are shown in Appendix I. We believe that such failure cases can largely be solved by introducing a visually aware summarization model, however, as of writing, no sufficiently large-scale general-purpose multi-modal model exists which is capable of serving this purpose.

**Controllability:** One of the key applications of image captioning systems is alt-text generation. As discussed in recent work (Archibald, 2023), alt-text generation is largely contextual, which means that for each image, the alt-text of the image should depend on the context that such an image is included in. While IC³ introduces a natural pathway for including context through the summarization model, we have found (see Appendix I.2), that IC³ is somewhat resistant to prompts that encourage surfacing background information. Exploring how to make IC³ surface arbitrary information in the image instead of focusing primarily on foreground information is key direction for future work.

**The Cost of using LLMs:** The use of many closed source large language models can represent a significant financial, human, and environmental cost (Bender et al., 2021). We recognize that for some researchers and students, the financial cost of using a large zero-shot model such as GPT-3 can be prohibitive, making IC³ difficult to compare against, especially for large-scale experiments such as the Karpathy test set for MS-COCO and the Flicker-30K datasets (which consists of 5K images each). Using GPT-3, IC3 costs about \$0.0109/Image, and with GPT-3.5, that cost falls to \$0.001/Image. Notably, this is significantly less than Chat Captioner (Zhu et al., 2023), which can cost as much as \$0.27/Image, which made it infeasible to run large-scale experiments. The experiments/ablations/all GPT-3 tuning in this paper was performed for \$250 (USD). Our approach, while not necessarily cheap, is several orders of magnitude less expensive than training/evaluating fine-tuned vision and language models such as Flamingo (1536 TPUs/15 days, roughly \$1,780,531 using on-demand TPU pricing) or BLIP-2 (16 A100 GPUs, 6 days, \$11,796 using AWS on-demand pricing). Furthermore, we hope that this cost will not be prohibitive long-term. GPT-3.5 is an order of magnitude cheaper, and has similar performance to GPT-3, and open-weight models such as Koala and Vicuna, seem promising for the future of affordable LLMs (see Appendix C.1), making IC3 even more accessible to students and researchers.

## 6  Conclusion

In this work, introduce IC³, a method for image captioning that first samples captions from multiple viewpoints and then summarizes and filters them to create high-fidelity descriptions. As far as we are aware, IC³ is the first work to demonstrate a pipeline for generating a single caption by integrating distributionally-faithful candidate captions, and does so without changing model architecture or retraining by leveraging summarization to produce a single omnibus caption capturing the full distribution of information. Further, IC³ is the first work for paragraph captioning or image captioning that uses summarization of distributionally-faithful caption samples and the first to demonstrate in human experiments that long-form captions encoding this distribution are preferable to single reference captions. Human users rate IC³ captions at least as helpful as baseline captions up to 80% of the time, and such IC³ captions are capable of inducing up to 84% relative improvements in recall approaches over baseline captioning methods. While our implementation of IC³ is relatively simple, it demonstrates significant gains over traditional paradigms, suggesting that this is only the beginning for caption sampling and summary methods.

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

## Appendix

## A    Acknowledgements

Authors, as part of their affiliation with UC Berkeley, were supported in part by the Berkeley Artificial Intelligence Research (BAIR) industrial alliance program.

## B    Code Release

Our code is available at `https://github.com/davidmchan/caption-by-committee`, and is made publicly available on Github with an MIT license, and contains the implementation, as well as the validation results for each of the models, the evaluation server/framework, and other necessary artifacts, to encourage further research in the domain of diverse/summarized image captioning.

## C    Hyperparameter Exploration

In this section, we provide additional experimental details regarding the choice of the hyperparameters for our method discussed in section 3.

### C.1    Choice of Summarization Model

The choice of the summarization model $\mathcal{S}$ is a key decision when implementing $IC^3$. Table A1 demonstrates the performance of $IC^3$ with several models, both using prompting for large language models, and using summarization of the captions directly. Generally we found that models from OpenAI (Such as GPT-3 and GPT-4) were strong performers, however models from Anthropic (such as CLAUDE), have strong summarization performance as well. The strongest open-source models are Koala and Vicuna, both Chat-style models, with LLama and StableLM following. While Table A1 seems to imply that T-5 is a strong model (and it likely is in terms of content-coverage in recall), T-5 often copy-pastes several of the candidate sentences instead of generating a strong abstractive summary, leading to decreased fluency.

### C.2    Prompts

In this section, we present several explorations of possible prompts. First, Table A2, we present an exploration of the prompt with and without language which encourages the model to produce uncertainty-specific language (the green text in subsection 3.4). To evaluate this, we use two approaches: a head-to-head experiment where captions generated by the two prompts are evaluated directly by human raters for helpfulness and correctness (following subsection 3.5), and an automated measure of "likely-language occurrence", LLOP. To compute LLOP, we compute the number of captions containing words that indicate some uncertainty including "likely", "probably", "possibly" and others. We find that without explicitly encouraging the model to produce uncertain language, the model seldom does so, while doing so improves both the helpfulness and correctness when measured by human raters.

In the second exploration in Table A3, we explore the question of using a prefix similar to one explored in Kojima et al. (2022). We find that while the prefix does help, it is not a key component of our method, and increases automated measures by small, but perceivable, levels.

### C.3    Choosing the number of candidate samples

Choosing the number of captions to summarize is highly dependent on both the abilities of the language model, and the tolerance for execution cost of the method. Adding more captions can increase

Table A1: Exploration of the choice of language model, when holding the prompt and candidate captions stable, using BLIP on a 200-element randomly sampled subset of the MS-COCO dataset.

| | EXACT | | FUZZY | | CLIP RECALL | | | |
|---|---|---|---|---|---|---|---|---|
| MODEL | NOUN | VERB | NOUN | VERB | MRR | R@1 | R@5 | R@10 |
| **LANGUAGE MODELS** | | | | | | | | |
| IC$^3$ + BLOOM (SCAO ET AL., 2022) | 0.248 | 0.16 | 0.551 | 0.402 | 0.834 | 0.725 | 0.98 | 0.995 |
| IC$^3$ + DISTILGPT2 (SANH ET AL., 2019) | 0.208 | 0.146 | 0.517 | 0.488 | 0.643 | 0.535 | 0.795 | 0.825 |
| IC$^3$ + GPT2 (RADFORD ET AL., 2019) | 0.272 | 0.159 | 0.602 | 0.542 | 0.638 | 0.51 | 0.79 | 0.83 |
| IC$^3$ + GPT2 LG (RADFORD ET AL., 2019) | 0.28 | 0.164 | 0.583 | 0.486 | 0.735 | 0.64 | 0.85 | 0.89 |
| IC$^3$ + GPT2 MED (RADFORD ET AL., 2019) | 0.299 | 0.187 | 0.606 | 0.531 | 0.79 | 0.705 | 0.9 | 0.935 |
| IC$^3$ + GPT2 XL (RADFORD ET AL., 2019) | 0.28 | 0.18 | 0.58 | 0.473 | 0.849 | 0.755 | 0.975 | 0.99 |
| IC$^3$ + GPT3 (BROWN ET AL., 2020) | | | | | | | | |
| + ADA | 0.282 | 0.18 | 0.585 | 0.463 | 0.866 | 0.78 | 0.975 | 0.985 |
| + BABBAGE | 0.199 | 0.115 | 0.504 | 0.341 | 0.83 | 0.735 | 0.97 | **1.0** |
| + CURIE | 0.218 | 0.111 | 0.519 | 0.319 | 0.827 | 0.71 | 0.975 | 0.995 |
| + DAVINCI2 | 0.321 | 0.207 | 0.622 | 0.491 | 0.939 | 0.895 | **1.0** | **1.0** |
| + DAVINCI3 | 0.381 | 0.251 | 0.675 | 0.547 | 0.958 | **0.925** | **1.0** | **1.0** |
| IC$^3$ + GPTNEO 125M (BLACK ET AL., 2021) | 0.235 | 0.157 | 0.521 | 0.447 | 0.777 | 0.69 | 0.895 | 0.915 |
| IC$^3$ + GPTNEO 1B (BLACK ET AL., 2021) | 0.253 | 0.155 | 0.546 | 0.403 | 0.844 | 0.75 | 0.985 | 0.995 |
| IC$^3$ + GPTNEO 2B (BLACK ET AL., 2021) | 0.242 | 0.15 | 0.536 | 0.393 | 0.844 | 0.74 | 0.98 | **1.0** |
| IC$^3$ + LLAMA7B (TOUVRON ET AL., 2023) | 0.224 | 0.128 | 0.517 | 0.324 | 0.777 | 0.65 | 0.945 | 0.995 |
| IC$^3$ + LLAMA13B (TOUVRON ET AL., 2023) | 0.257 | 0.175 | 0.554 | 0.419 | 0.834 | 0.725 | 0.99 | **1.0** |
| IC$^3$ + STABLE LM 3B (STABILITYAI, 2023) | 0.247 | 0.184 | 0.552 | 0.454 | 0.873 | 0.785 | 0.985 | 0.995 |
| **CHAT MODELS** | | | | | | | | |
| IC$^3$ + ALPACA 7B (TAORI ET AL., 2023) | 0.324 | 0.216 | 0.63 | 0.503 | 0.912 | 0.85 | **1.0** | **1.0** |
| IC$^3$ + CHATGPT (OPENAI, 2022) | 0.401 | 0.27 | 0.692 | 0.595 | 0.954 | 0.920 | **1.0** | **1.0** |
| IC$^3$ + CLAUDE (BAI ET AL., 2022) | 0.38 | 0.262 | 0.677 | 0.583 | **0.962** | **0.935** | **1.0** | **1.0** |
| IC$^3$ + GPT4 (OPENAI, 2023) | **0.42** | 0.29 | **0.713** | 0.609 | 0.96 | 0.925 | **1.0** | **1.0** |
| IC$^3$ + KOALA 7B (GENG ET AL., 2023) | 0.284 | 0.178 | 0.586 | 0.455 | 0.899 | 0.825 | 0.99 | **1.0** |
| IC$^3$ + KOALA 13B v1 (GENG ET AL., 2023) | 0.418 | **0.323** | 0.692 | **0.637** | 0.916 | 0.865 | 0.985 | 0.985 |
| IC$^3$ + KOALA 13B v2 (GENG ET AL., 2023) | 0.376 | 0.264 | 0.67 | 0.592 | 0.923 | 0.87 | 0.995 | 0.995 |
| IC$^3$ + STABLE LM 3B (STABILITYAI, 2023) | 0.077 | 0.06 | 0.299 | 0.144 | 0.265 | 0.23 | 0.28 | 0.295 |
| IC$^3$ + STABLE LM 7B (STABILITYAI, 2023) | 0.263 | 0.197 | 0.564 | 0.489 | 0.873 | 0.785 | 0.995 | **1.0** |
| IC$^3$ + VICUNA 7B (CHIANG ET AL., 2023) | 0.361 | 0.247 | 0.658 | 0.548 | 0.938 | 0.89 | **1.0** | **1.0** |
| IC$^3$ + VICUNA 13B (CHIANG ET AL., 2023) | 0.384 | 0.272 | 0.676 | 0.584 | 0.927 | 0.87 | 0.995 | 0.995 |
| **SUMMARY MODELS** | | | | | | | | |
| IC$^3$ + T5 BASE (RAFFEL ET AL., 2020) | 0.353 | 0.25 | 0.646 | 0.587 | 0.903 | 0.845 | 0.98 | 0.99 |
| IC$^3$ + T5 SMALL (RAFFEL ET AL., 2020) | 0.402 | 0.289 | 0.681 | 0.609 | 0.944 | 0.9 | 0.995 | **1.0** |
| **BASELINES** | | | | | | | | |
| IC$^3$ + REFERENCES | 0.434 | 0.305 | 0.684 | 0.564 | 0.939 | 0.895 | 0.995 | 1.0 |
| BLIP BASELINE (LI ET AL., 2022) | 0.266 | 0.196 | 0.567 | 0.491 | 0.865 | 0.77 | 0.995 | 1.0 |
| CHAT CAPTIONER (T5-XXL + CHATGPT) (ZHU ET AL., 2023) | 0.361 | 0.207 | 0.669 | 0.564 | 0.947 | 0.905 | **1.0** | **1.0** |

Table A2: Exploration of "uncertainty-encouraging" language in the prompt, using BLIP and GPT-3 on a 200 element randomly sampled subset of the MS-COCO dataset. See Appendix C.2 for a discussion of LLOP, the "likely-language occurrence percentage". Helpfulness and correctness are given as head-to-head win percentage following subsection 3.5.

| MODEL | LLOP | HELPFULNESS | CORRECTNESS |
|---|---|---|---|
| CANDIDATES WITH | **62.5%** | **52.01%** | **72.32%** |
| CANDIDATES WITHOUT | 4.0% | 43.63% | 18.16% |
| REFERENCES WITH | **52.0%** | **34.65%** | **53.00%** |
| REFERENCES WITHOUT | 0% | 28.79% | 26.20% |

Table A3: Content coverage and CLIP recall demonstrating the use of "This is a hard problem" in the prompt, using BLIP on a 200 element randomly sampled subset of the MS-COCO dataset.

| MODEL | EXACT NOUN | VERB | FUZZY NOUN | VERB | CLIP RECALL MRR | R@1 | R@5 | R@10 |
|---|---|---|---|---|---|---|---|---|
| CANDIDATES WITH | **0.322** | **0.216** | **0.647** | **0.503** | **0.770** | **0.645** | **0.96** | **0.99** |
| CANDIDATES WITHOUT | 0.316 | 0.208 | 0.638 | 0.496 | 0.765 | 0.635 | 0.94 | **0.99** |
| REFERENCES WITH | 0.516 | **0.308** | 0.744 | **0.560** | **0.833** | **0.745** | **0.97** | **0.995** |
| REFERENCES WITHOUT | **0.518** | 0.305 | **0.745** | 0.558 | 0.830 | **0.745** | **0.97** | 0.99 |

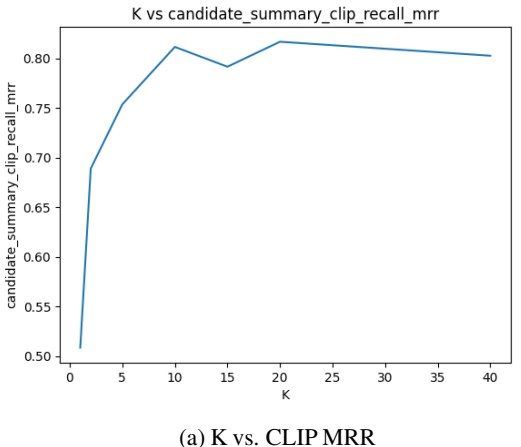

(a) K vs. CLIP MRR

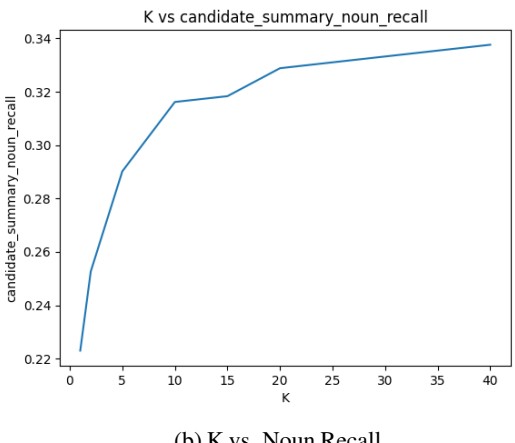

(b) K vs. Noun Recall

Figure A1: Exploration of the number of candidate set captions vs CLIP MRR and Noun Recall for the GPT-3 language model.

the amount of information discovered by the visual model, and generate better representative samples of the input data distribution, however it can increase the context length passed to the large language model, straining the summarization capabilities of the model, and leading to increased cost for the LLM. We ablate the choice in Figure A1. Here, we can see that increasing the number of captions can lead to increases in automated measure performance (as more captions will capture more information), however more captions can also increase execution time linearly with the number of candidate captions. We can see here that much of the benefit is captured at 10 candidate captions, which we chose for this work, since it represents a good trade-off between execution time, and caption quality.

**Is performance just due to LLMs correcting caption errors?** Table Table A4 shows both the automated performance of the GPT-3 model, and the human performance of the GPT-3 model for several values of K. We can see that while GPT can help to improve on single captions through error correction (as evidenced by slightly higher scores for GPT-3 (K=1) vs. the baseline), the best scores are achieved with higher values of K.

### C.4 BLIP + OFA

Because the caption summarization process is independent of the caption generation process, it is a natural question to ask if multiple different sources of caption generation could be used during the generation phase. The results of combining the sampled candidates from both BLIP and OFA are shown in Table A5

### C.5 How is caption diversity related to IC3 outputs?

One reasonable question to ask is: does the diversity of the input captions impact the quality of the output summarized caption? In Figure A2, we plot the Self-BLEU (Zhu et al., 2018) of the candidate captions (a measure of caption-set diversity), against the automated evaluation measures. We find that in general, there are very weak correlations between the CLIP MRR and the diversity of the candidate set (OFA, $r = 0.079$, BLIP, $r = 0.094$, BLIP-2, $r = 0.059$): when more diversity is needed to express the content to high specificity, the model is including it. When less diversity is required, the model does not include it. We do however see correlation between the content recall scores of the model, and the diversity

Table A4: Exploration of the choice of K for the GPT-3 language model and the BLIP-2 captioning engine on a randomly sampled 200 element MS-COCO subset. Human results are given as Glicko-2 scores (See Appendix E).

| K | EXACT | | FUZZY | | CLIP RECALL | | | HUMAN |
|---|---|---|---|---|---|---|---|---|
| | NOUN | VERB | NOUN | VERB | MRR | R@1 | R@5 | GLICKO |
| BASELINE | 0.264 | 0.162 | 0.564 | 0.423 | 0.885 | 0.805 | 0.985 | 1367.28 |
| 1 | 0.258 | 0.161 | 0.562 | 0.456 | 0.872 | 0.790 | 0.985 | 1534.48 |
| 10 | 0.346 | 0.212 | 0.646 | 0.516 | **0.956** | **0.920** | **1.000** | 1674.51 |
| 100 | **0.368** | **0.223** | **0.665** | **0.526** | 0.948 | 0.905 | **1.000** | 1623.22 |

Table A5: Content coverage and CLIP recall demonstrating the combination of caption engines on a 200 element randomly sampled subset of the MS-COCO dataset.

| MODEL | EXACT | | FUZZY | | CLIP RECALL | | | |
|---|---|---|---|---|---|---|---|---|
| | NOUN | VERB | NOUN | VERB | MRR | R@1 | R@5 | R@10 |
| OFA + BLIP + IC[3] | 0.341 | 0.204 | 0.648 | 0.485 | 0.796 | 0.685 | 0.945 | 0.985 |
| REFS + IC[3] | 0.517 | 0.308 | 0.744 | 0.561 | 0.833 | 0.745 | 0.970 | 0.995 |
| BLIP + IC[3] | 0.313 | 0.206 | 0.637 | 0.493 | 0.770 | 0.645 | 0.960 | 0.99 |
| OFA + IC[3] | 0.300 | 0.184 | 0.623 | 0.474 | 0.770 | 0.660 | 0.935 | 0.97 |
| BLIP | 0.230 | 0.178 | 0.542 | 0.439 | 0.551 | 0.400 | 0.760 | 0.91 |
| OFA | 0.212 | 0.150 | 0.531 | 0.387 | 0.341 | 0.115 | 0.630 | 0.89 |
| REFS | 0.214 | 0.537 | 0.099 | 0.337 | 0.683 | 0.540 | 0.877 | 0.965 |

of the input candidates (Noun Recall: OFA, $r = 0.233$, BLIP $r = 0.238$, BLIP-2, $r = 0.252$, Verb Recall: OFA, $r = 0.199$, BLIP $r = 0.193$, BLIP-2, $r = 0.185$). This suggests that when the candidates are more diverse, this information is captured in the output summary sentence.

# D Additional Experimental Details

## D.1 Image Captioning Methods

In this work, we explore two image captioning models as seed models for IC[3]: BLIP (Li et al., 2022) and OFA (Wang et al., 2022b).

**BLIP:** BLIP (Li et al., 2022) is a vision-language pre-training framework designed to effectively use noisy web data at scale for pre-training. The model operates by using a large dataset of synthetic image-caption pairs, generated by a seed captioning model, and a filter to remove low-quality synthetic captions. BLIP has demonstrated strong transfer performance to many vision-language tasks, and performs particularly well when transferred to image captioning in zero-shot and fine-tuning scenarios. The BLIP (Large) model that we use is fine-tuned on MS-COCO for image captioning, and unless otherwise specified, we generate baseline captions using beam search with 16 beams, and generate candidate captions for IC3 using temperature sampling as described in subsection 3.2.

**OFA:** OFA (Wang et al., 2022b) is a unified paradigm for multimodal pre-training, which is both task and modality agnostic. For pre-training, OFA unifies several vision and language tasks including image generation, visual grounding, image captioning, image classification and language modeling among others, and is pre-trained using 20M publicly available image-text pairs. The OFA (Huge) model that we use is fine-tuned on MS-COCO for image captioning, and unless otherwise specified, we generate baseline captions using beam search with 16 beams, and generate candidate captions for IC3 using temperature sampling as described in subsection 3.2.

## D.2 Datasets

We explore image captioning across several datasets in this paper.

**MS-COCO Dataset:** The MS-COCO dataset (Lin et al., 2014) is a dataset for image description containing 328K images, each with 5 ground truth descriptions. MS-COCO is licensed under a Creative Commons Attribution 4.0 license. All of the results in this work are presented on the test-split of the Karpathy splits of the COCO-2014 dataset.

**Flickr-30K Dataset:** The Flickr-30K dataset (Young et al., 2014) is an image description dataset containing 30K images and 150K captions (5

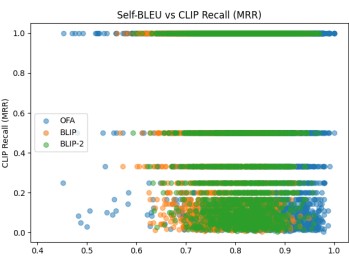 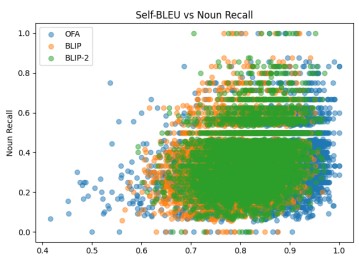 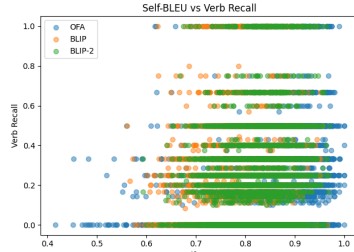

Figure A2: Plots showing diversity of candidate captions plotted against automated evaluation measures when using 10 candidate captions, and GPT-3 (Davinci v3) as a LM.

ground truth captions per image), and is licensed under a custom non-commercial (for research use only) license. All of the results are presented on the test-split of the Karpathy splits of the Flickr-30K dataset.

**Hard MRR Splits:** In some situations, we want to be able to explore the performance of our model vs. baselines on the most challenging captions. We call these splits the "Hard MRR" splits, and they consist of the set of 200 samples for which the MRR of the underlying captioning model is lowest. Thus, HARD MRR - BLIP contains the 200 samples minimizing MRR for the baseline BLIP model, with a caption generated using beam search (16 beams), and similarly HARD MRR - OFA contains the 200 samples minimizing MRR for the baseline OFA model with a caption generated using beam search (16 beams).

### D.3 N-Gram Metric Scores

The performance of the model on traditional n-gram measures is demonstrated in Table A6. In this work, IC$^3$ models are designed to produce captions which are the combination of all of the viewpoints presented by each of the individual captions, suggesting that they contain more information on average than any single reference sentence. Because of this, often the n-gram performance of the model is significantly worse, as while the overlap of content n-grams may be higher (suggested by Table 7 and Table 8 in the main paper), there are a lot of extra n-grams per caption, which will decrease metric scores. We explore four n-gram measure: BLEU (Papineni et al., 2002), CIDEr: (Vedantam et al., 2015), METEOR (Agarwal and Lavie, 2008) and ROUGE-L (Lin, 2004). We also compute the MAUVE (Pillutla et al., 2021) score between *all* samples generated and the reference samples, which measures the deviation in the language space, and notice that the MAUVE score is extremely low, suggesting that we have succeeded in producing a language distribution which is significantly

different from the reference distribution.

## E ELO Scoring for Human Ratings

The results shown in subsection 4.1 indicate a challenging reality: humans can often find it difficult to calibrate to the quality of image captions when viewing single image captions alone, but can find it much easier to understand any differences in quality when presented with two pairs of captions, in a head to head fashion. Unfortunately, since human caption ratings can be expensive, it is tricky to perform all head to head caption evaluations across values of K, language models, captions, etc. In order to compensate for this, in some situations instead of running the full head to head experiment, we instead use a tournament, which measures the quality of a model through an Glicko-2 score-based rating system (Glickman, 1995). In some cases, we report the Glicko-2 scores of each of the models in our human-rating tournament, as a proxy for the overall quality of the model.

## F Human Studies

In our work, we run two different human rating studies, a head-to-head comparison between methods, and a context-free method which generates mean opinion scores. A screenshot of our evaluation tool for mean opinion scores is given in Figure A11, and a screenshot of the evaluation tool for head-to-head rating is given in Figure A12. Both of these experiments have been approved as *exempt* under category 2 by the OMITTED-FOR-REVIEW IRB, protocol ID 2022-11-15846. For any questions, contact OMITTED-FOR-REVIEW .

Significant prior work has explored the collection of human judgments of the quality of visual descriptions. Human judgment is considered the gold standard for visual description evaluation, and previous studies typically rely on human annotators to rate caption quality on one or multiple

Table A6: Performance of models augmented with IC$^3$ on traditional N-gram measures.

| MODEL | BLEU@4 ↑ | CIDER ↑ | ROUGE-L ↑ | MAUVE ↑ |
|---|---|---|---|---|
| **MS-COCO (KARPATHY SPLIT)** | | | | |
| OFA + IC$^3$ | 0.159 | 0.495 | 0.483 | 0.091 |
| BLIP + IC$^3$ | 0.118 | 0.325 | 0.445 | 0.074 |
| OFA | 0.292 | 1.323 | 0.598 | 0.254 |
| BLIP | 0.292 | 1.315 | 0.595 | 0.158 |
| **FLICKR-30K (TEST SPLIT)** | | | | |
| OFA + IC$^3$ | 0.132 | 0.392 | 0.449 | 0.004 |
| BLIP + IC$^3$ | 0.092 | 0.277 | 0.401 | 0.004 |
| OFA | 0.212 | 0.872 | 0.541 | 0.004 |
| BLIP | 0.160 | 0.501 | 0.727 | 0.004 |

axes (Levinboim et al., 2021; Kasai et al., 2022). While automated methods exist for the evaluation of caption quality (Agarwal and Lavie, 2008; Vedantam et al., 2015; Papineni et al., 2002), recent work including THUMB (Kasai et al., 2022), which has run human evaluations on captions produced by models based on "Precision", "Recall", "Fluency", "Conciseness" and "Inclusive Language", has shown that humans produce captions which score significantly higher when judged by human raters, than when judged by existing measures (and further, that human judgments of quality correlate poorly with existing measures), necessitating the need for human evaluation as opposed to evaluation of captioning methods using automated measures for caption quality. Our model quality evaluation method is closely aligned with the work in (Levinboim et al., 2021), where we use similar questions to determine the "helpfulness" and "correctness" of visual descriptions. Our work differs in that instead of collecting a set of human ratings of captions for the purpose of training statistical models, we aim to evaluate the quality of both human and machine generated captions, in an effort to determine if the machine generated captions from recently proposed methods in our group outperform existing human and machine generated captions for the images.

In this study, subjects participate in sessions of reviewing visual media with corresponding visual descriptions, e.g. pairs of images and captions. These sessions consist of sequences of rating tasks, with a session consisting of not more than ten tasks. The types of tasks, which we call activities, are as follows:

- Caption Rating - Viewing a given image and caption pair, and rating the quality of the caption on several axes (described below).

- head-to-head Caption Rating - Viewing a given image, and a pair of captions, and deciding which caption better describes the image.

While there are several possible activities, each session consists of sequences of the same type of activity, and each task is presented in randomized order for each subject.

Subjects are linked to the data collection interface on our server developed by us in a frame directly from an Amazon Mechanical Turk internal HIT using the ExternalQuestion API which allows external web content to be displayed within the internal HIT. No third-party software is used with the HITs and no reviewing data is collected by Amazon or any third-parties with the use of this API.

The subjects are shown a consent form on the Amazon Mechanical Turk HIT prior to entering our data collection interface. Subjects are then required to click the "I Accept" button to confirm their agreement with the consent information of the study. They are then redirected to the data collection interface. For each image, users are presented with an image, and an associated image description. Images are drawn from the MSCOCO dataset (Lin et al., 2014). Human generated captions are drawn from the references collected by the authors of (Lin et al., 2014).

For task A (Caption Rating), users are first asked to rate the "helpfulness" of the caption, with the prompt: "Does the caption provide useful information for a person who cannot see the image (and who is trying to understand what is in the image)?", among the options: Not helpful at all", "Slightly helpful", "Moderately helpful", "Helpful", "Very helpful".. The user can also select the option "I can't tell". The user is then asked to rate the "Correctness" of the caption with the prompt "How

correct is the caption?", among the options "Completely wrong", "Mostly wrong" "Slightly wrong", "Slightly right", "Mostly right", "Completely right". The user can also select the option "I can't tell".

The user is then asked to select the "submit" button, to move to the next task in the HIT, which is composed of 10 tasks. The user can also skip the task by selecting the "Image/Caption not visible button". If the user selects "I can't tell" or "Image/Caption not visible" for any option, the tasks remaining are not decreased, but if the user selects submit, and passes a valid rating, then the number of tasks remaining are reduced by 1.

For task B (head-to-head Caption Rating), the user is presented with two image captions instead of one image caption, and asked to select "Caption A better represents the content in the image" or "Caption B better represents the image". The user can also select "I can't tell". The user is then asked to select the "submit" button, to move to the next task in the HIT, which is composed of 10 tasks. The user can also skip the task by selecting the "Image/-Caption not visible button". If the user selects "I can't tell" or "Image/Caption not visible" for any option, the tasks remaining is not decreased, but if the user selects submit, and passes a valid rating, then the number of tasks remaining is reduced by 1.

After completing all of the tasks in the session, users are given a randomly generated code, which is entered in the Amazon MTurk HIT page, and links the user's survey results to the Amazon worker ID. We collect these linkings to perform analysis on inter-rater agreement, as while the session itself is anonymous, users may complete multiple sessions, and some method is required to maintain identity between the sessions.

After each of these sessions, subjects are given a brief survey regarding the task difficulty (Select from the options: "Very Easy", "Easy", "Normal", "Hard", "Very Hard") and prompted for any additional comments on the session in general for each session in an (optional) open-response format. Users are also encouraged to protect their privacy with the prompt: "After submitting your responses, you can protect your privacy by clearing your browser's history, cache, cookies, and other browsing data. (Warning: This will log you out of online services.)" Subjects were compensated with $0.18 USD per session (based on the recommended Amazon wage (federal minimum wage, $7.25/Hr),

with an expected completion time of 1.5 minutes per session), and should be able to complete the session in under one and half minutes (based on several pilot examples). Subjects can participate in the task a maximum of 100 times. The maximum time commitment for each subject over two months of our study is 2 hours.

To compute p-values, we first aggregate each users' session scores for each model (in the case of MOS, we take the mean for each model, and in the case of head-to-head, we assign a $+1$ value for a win, and a $-1$ value for a loss, and take the mean). For MOS, we compute a 1-sided t-test on the aggregated samples (which should be independent) to the baseline, while for the head-to-head scores, we compute a 1-sided single-sample t-test against a mean of zero.

# G  Additional Qualitative Examples

Additional qualitative examples are given in Figure A3, Figure A4, Figure A5, Figure A6 and Figure A7. From these examples, it is clear that IC$^3$ outperforms the baseline in many situations. Examples in this section are randomly selected from the test set when indicated.

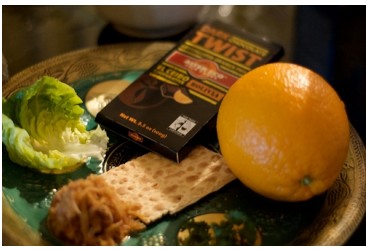 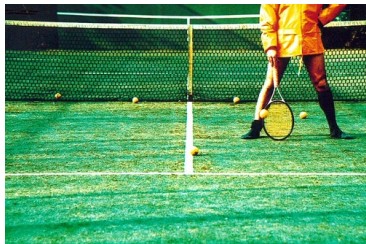 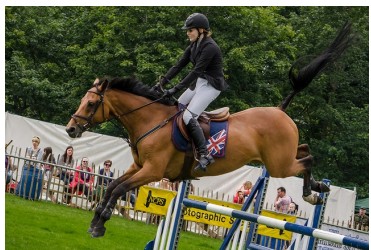

(a) BLIP+IC³: A plate with an orange, crackers, lettuce, and possibly other items such as nuts or a book.
BLIP: A close up of a plate of food on a table.

(b) BLIP+IC³: A man standing on a tennis court holding a tennis racquet, possibly wearing an orange outfit or raincoat.
BLIP: A man standing on top of a tennis court holding a racquet.

(c) BLIP+IC³: A woman riding a brown horse and jumping over hurdles in a competition, with other people watching.
BLIP: A woman riding on the back of a brown horse.

Figure A3: Additional qualitative examples of BLIP + IC³ on the MS-COCO dataset.

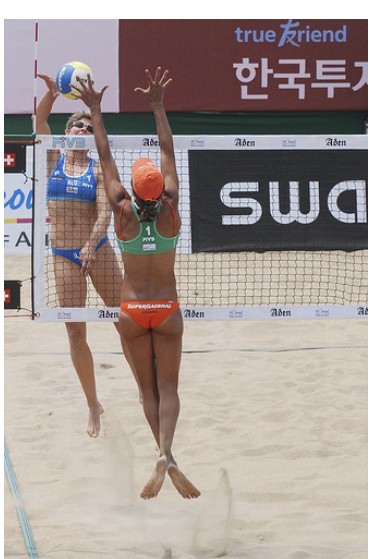 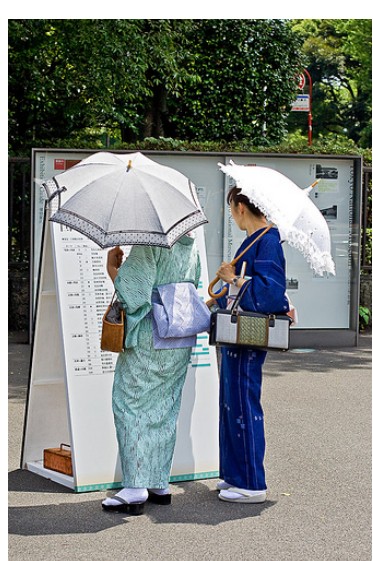 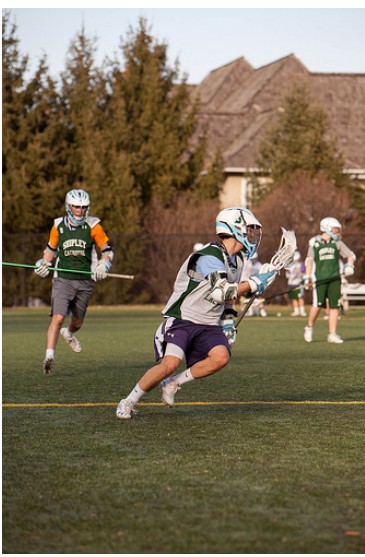

(a) OFA+IC³: A woman in a bikini jumping in the air to hit a volleyball on a beach, possibly while playing a game of beach volleyball.
OFA: A woman in a bikini is jumping in the air to hit a volleyball.

(b) OFA+IC³: Two women in kimonos standing in front of an information board with umbrellas, possibly in the rain.
OFA: Two women with umbrellas standing in front of an information board.

(c) OFA+IC³: A lacrosse player in a white jersey running down the field with the ball during a game or match.
OFA: A lacrosse player runs with the ball.

Figure A4: Randomly selected qualitative examples of OFA + IC³ on the Flickr30K dataset.

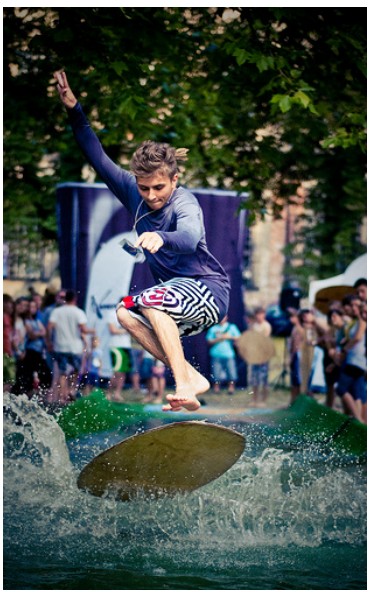
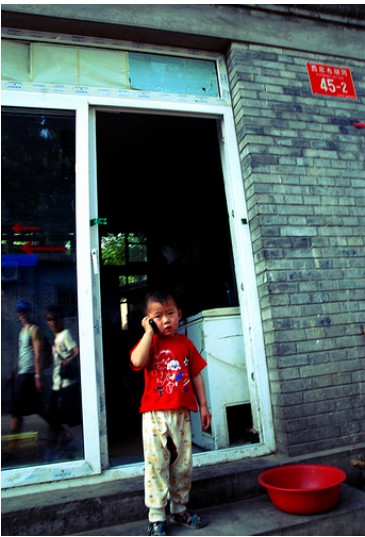
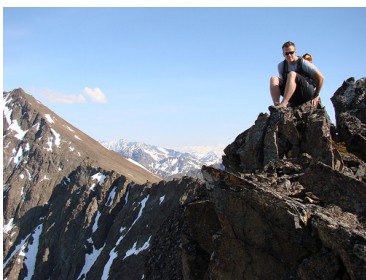

(a) BLIP+IC³: A man in striped trunks riding a surfboard on a large wave near a group of people.
BLIP: A man riding a wave on top of a surfboard.

(b) BLIP+IC³: A young boy standing outside of a building, possibly in front of a window or doorway, holding a cell phone to his ear and wearing a red shirt.
BLIP: A little boy standing outside of a building talking on a cell phone.

(c) BLIP+IC³: Two people, possibly hikers, sitting on top of a mountain, possibly icy or rocky, overlooking a snowy valley.
BLIP: A couple of people sitting on top of a mountain.

Figure A5: Randomly selected qualitative examples of BLIP + IC³ on the Flickr30K dataset.

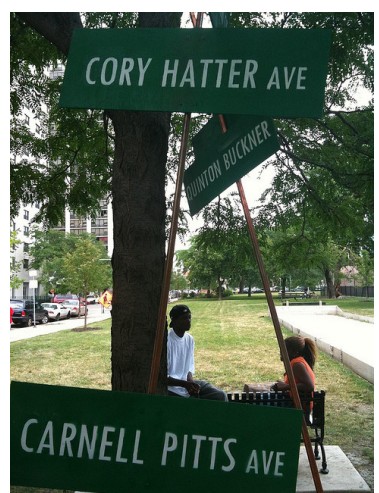
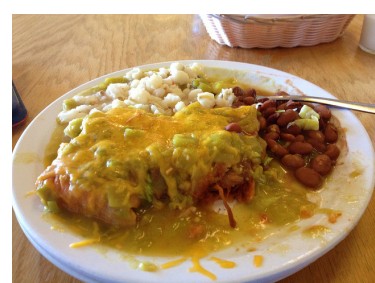
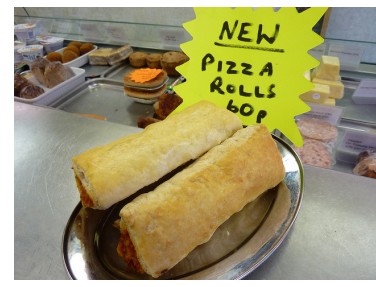

(a) OFA+IC³: A group of people sitting on a bench under a tree, with four green street signs hanging from it.
OFA: A group of people sitting on a bench under a tree.

(b) OFA+IC³: A plate of food on a table with rice, beans, and possibly a meat dish, such as chicken or mashed potatoes.
OFA: A plate of food on a table.

(c) OFA+IC³: A plate with two items of food on it, possibly a sandwich and a burrito, or two empanadas, sitting on a table or counter.
OFA: A plate of food on a table.

Figure A6: Randomly selected qualitative examples of OFA + IC³ on the MS-COCO dataset.

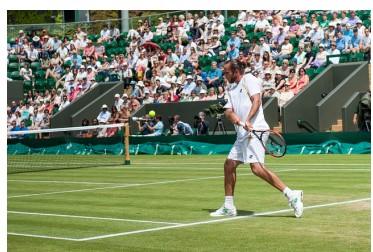 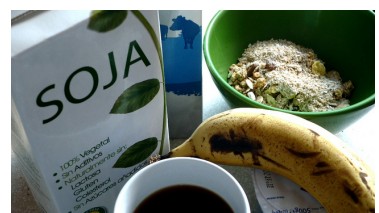 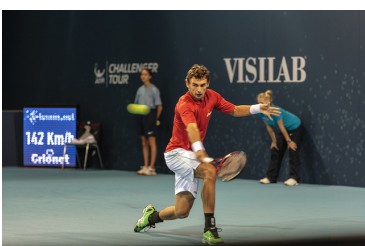

(a) BLIP+IC³: A male tennis player wearing all white, walking across a tennis court while holding a racquet, possibly after losing a match.
BLIP: A man walking across a tennis court holding a racquet.

(b) BLIP+IC³: A banana, bowl of cereal, and cup of coffee sitting on a table or counter.
BLIP: A banana sitting next to a bowl of cereal and a cup of coffee.

(c) BLIP+IC³: A man wearing either red or green and white, holding a tennis racquet and swinging it at a tennis ball on a tennis court.
BLIP: A man holding a tennis racquet on a tennis court.

Figure A7: Randomly selected qualitative examples of BLIP + IC³ on the MS-COCO dataset.

## H  Zero-Shot Style and Language Transfer

It is well known that models such as GPT 3 (Radford et al., 2021) are capable of many zero-shot tasks, such as language style transfer and translation. By modifying the prompt in the summarization approach, $IC^3$ can be used to generate captions in different styles and languages. For example, we can modify the prompt to generate captions in different languages, for example, to generate captions in Japanese, we could use the prompt:

```
This is a hard problem.  Carefully
summarize in Japanese in ONE detailed
sentence the following captions by
different (possibly incorrect) people
describing the same scene.  Be sure
to describe everything, and identify
when you're not sure.  For example:
Captions:  {formatted captions}.
Summary (in Japanese): 写真はおそら
く
```

We can see the performance of the model for such prompts in Figure A8. Such captions represent an easy way to transfer knowledge to different languages, however may not outperform translating the English caption alone.

## I  Failure modes & Limitations

In this section of the appendix, we explore some of the limitations of the method, and provide some insight into how the model could be improved.

### I.1  Hallucination

Some examples of failure cases are shown in Figure A9. In Figure A9a, we can see the effect of "4th-wall breaking," one of the key failure modes of the method. Because the prompt suggests that that the model should combine several underlying captions, the output caption references the fact of this combination in a hidden way, when it says "other details varying". In some cases, the model might produce captions that end with words such as "... as stated by the captions" or "... but the captions differ." which both reference the prompt, and interfere with the flow of the caption.

In Figure A9b, we can see a situation where the model passes through a hallucination from the underlying captioning model. Because 3 of the 10 captions in the candidate set $K$ mention a cat: "A cat in a bathroom staring into a sink", "A bathroom with a toilet and sink and a cat", and "A cat walking around in a bathroom", and the LLM is not visually aware, there is no reason to doubt the existence of the cat, and it is included in the caption. Luckily, in this failure case the model prefaces the existence of the cat with a "may", however there are situations where this is not the case.

In Figure A9c, we can see the third major failure case of the model: treating uncertainty as multiple objects. Because the captioning model is not aware of the visual content of the image, when there is a high amount of noise in the captions, such as here, where the actual contents of the plate are unclear, the model often ascribes the noise to several objects in the scene, instead of a single uncertain object. This can sometimes be automatically detected by counting the number of commas in the caption, and we have found empirically that re-generating any caption with more than 7 commas can reduce or eliminate these effects (though we do not use this post-processing step in the paper).

### I.2  Controllable Alt-Text Generation

While our model is capable of generating high-fidelity descriptions of the image, as discussed in section 5, the model can struggle when asked to describe background and contextual details that differ significantly from the reference dataset distribution. To demonstrate this, we perform a case study with the image in Figure A10.

In the case study, we take the prompt:

```
This is a hard problem.  Carefully
summarize in ONE detailed sentence
the following captions by different
(possibly incorrect) people describing
the same scene.  Be sure to describe
everything, and identify when you're
not sure.  For example:  Captions:
{formatted captions}.  Summary:  I'm
not sure, but the image is likely
of...
```

and replace the colored prompt with a set of different prompts, to generate potential alt-text for the image in Figure A10. From these prompts, we can see that in many cases, the model fails to be controllable, and only repeats the key information in the scene. While in some cases the prompts can elicit additional information, we believe there is significant work to be done to explore how we can sample enough relevant information from the base

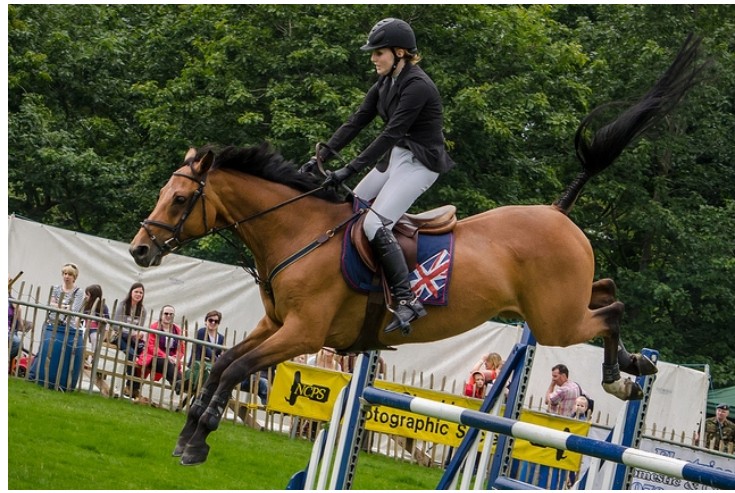

Figure A8: Examples of the generated caption (BLIP + IC$^3$) for different languages:

ENGLISH: A woman riding a brown horse and jumping over hurdles in a com- petition, with other people watching.

SPANISH: Una mujer montando a caballo un caballo marrón mientras salta un obstáculo en un campo verde, posiblemente en una competición con espectadores mirando.

FRENCH: Quelqu'un qui monte à cheval sur le dos d'un cheval brun et saute par-dessus un obstacle dans un champ, avec des gens en arrière-plan.

JAPANESE: 女性が茶色の馬の背中に乗って障害物を跳び越える様子を捉えたものであるが、確実ではない。

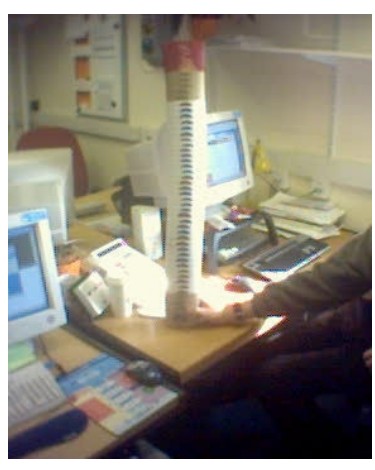

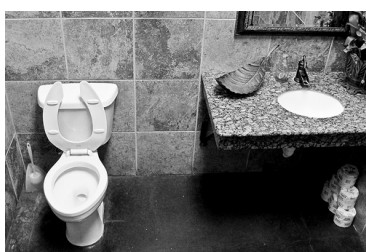

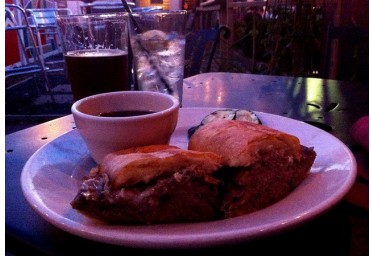

(a) Generated BLIP+IC$^3$ Caption: A man sitting at a desk in front of at least one computer, possibly two, **with other details such as clothing and accessories varying**.

(b) Generated BLIP+IC$^3$ Caption: A small bathroom with a white toilet, sink, counter, and possibly a marble tile floor, **and there may be a cat present**.

(c) Generated BLIP+IC$^3$ Caption: A white plate topped with a sandwich and a cup of coffee, **possibly accompanied by other food items such as french toast and/or meat**.

Figure A9: Examples of BLIP + IC$^3$ failure modes (MS-COCO Dataset).

captioning model (it seems that increasing the number of samples is not always sufficient), and surface that information in the generated summary (simple modifications to the prompt seem ineffective).

The results for BLIP are below:

**Prompt:** With original prompt

**Result:** A woman riding a bicycle past a building with a cafe, bar, hotel, and/or restaurant, possibly with a fence nearby.

**Prompt:** "Focus on describing what the person is doing"

**Result:** A person riding a bicycle past a building or establishment of some kind.

**Prompt:** "Focus on describing the buildings"

**Result:** A woman riding a bicycle past some buildings, possibly a hotel, with a gate in the background.

**Prompt:** "Only describe the buildings"

**Result:** A person riding a bicycle past a tall building.

**Prompt:** "do not mention any people in the summary"

**Result:** A person riding a bicycle down a street past buildings.

**Prompt:** "Focus on describing the color and architecture of the buildings"

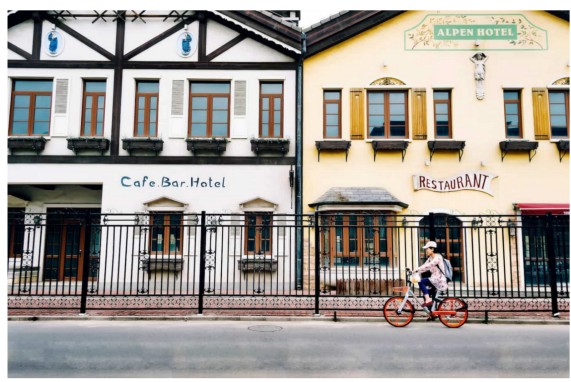

Figure A10: Photo by Zhang Kaiyv[1]. Alt-text is often contextual. From the reference: "If [the image] is used in an article about a public bicycle hire scheme named Hire-a-bike, then the bike is the focus, and the alt text could be "A woman rides a Hire-a-bike along a city road". However, if it's used in an article about a dispute between the café and the restaurant, the alt text might be "The storefronts of the 'Café Bar Hotel' and 'Alpen Hotel Restaurant'". Although if the dispute is about color of the buildings, the guttering, or the window frames, then those details should be part of the alt text."

**Result:** A person riding a bike past a hotel with colorful wooden shutters and other buildings nearby.

**Prompt:** "Focus on describing what the woman is wearing"

**Result:** A woman wearing casual clothing riding a bike past two colorful buildings.

**Prompt:** "Make sure to describe the appearance of the buildings in detail"

**Result:** A person riding a bicycle in front of a building, which could be an old hotel, cafe, apartment building, restaurant, or other type of building with many windows and possibly a fence or gate.

**Prompt:** "Make sure to describe the person's appearance in detail"

**Result:** A person, possibly a woman, riding a bicycle past buildings, a fence, a restaurant, and a hotel.

**Prompt:** "Make sure to describe the person's appearance in detail, such as what they are wearing and doing"

**Result:** A woman in a floral dress riding a red bike past an old, historic building with a restaurant across the street, while an older man is riding a bicycle past several other older-fashioned buildings. The results for OFA are below:

**Prompt:** With original prompt

**Result:** A woman riding a bicycle in front of a building, fence, or restaurant in a variety of cities in Europe.

**Prompt:** "do not mention any people in the summary"

**Result:** A person riding a bicycle in front of a building, fence, or restaurant.

**Prompt:** "Focus on describing what the person is doing"

**Result:** A person riding a bicycle in front of a building, fence, or restaurant.

**Prompt:** "Focus on describing what the person is doing" + "I'm not sure, but the image is likely of a person, who is"

**Result:** Possibly a woman, riding a bicycle on a street in front of a building, fence, metal gate, restaurant, hotel, or other structures in various cities such as Tallinn, Vienna, Albany, Alphen aan den Rijn, Berlin, Strasbourg, or Frankfurt am Main.

**Prompt:** "Make sure to include details about what the person is wearing and doing"

**Result:** A woman riding her bicycle past a tall fence, possibly with a restaurant behind it, in front of a building or hotel in a city such as Kitzingen, Albéliyah, Breckenridge, Frankfurt, Innbruck, or Bad Schwartau..

**Prompt:** "Make sure to only include details about what the person is wearing and doing"

**Result:** A woman riding a bicycle on a city street in front of a building, fence, or restaurant.

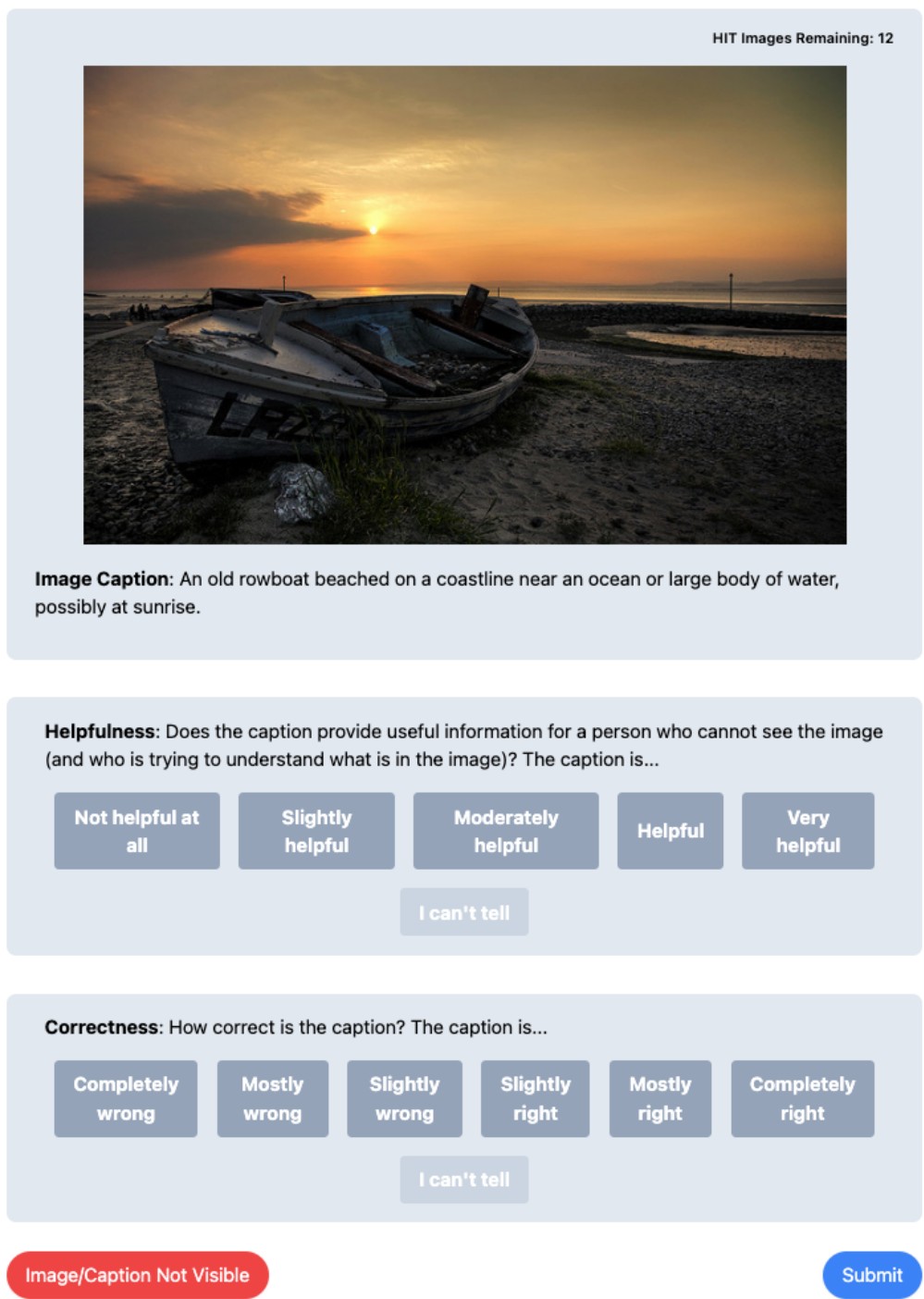

Figure A11: Description rating tool (Mean Opinion Scores).

# Description Rating Tool

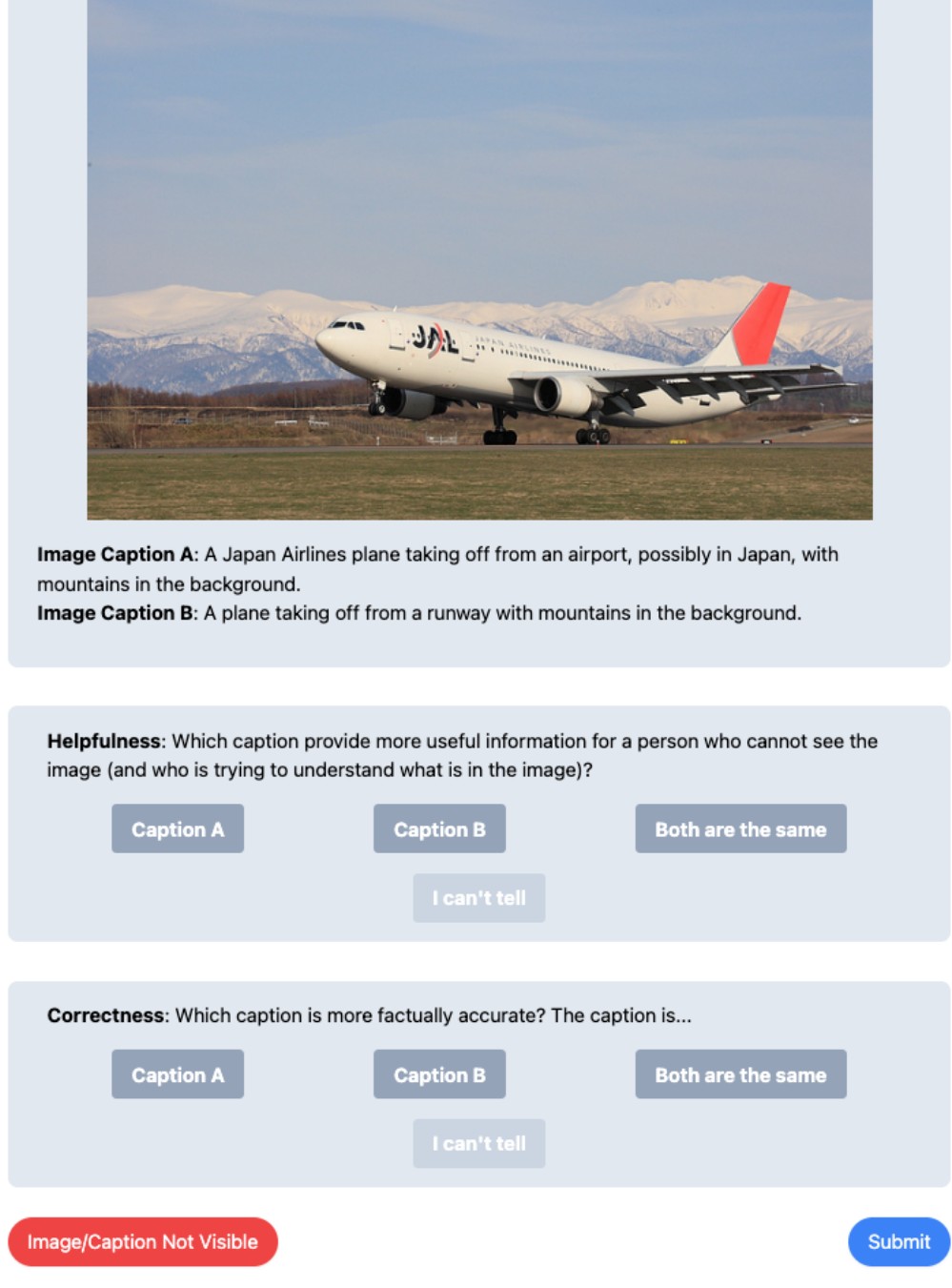

**Instructions:** Imagine that we would like to write a text description of the image below so that somebody who cannot see the image can understand the scene. Look at the image and the description, then answer the questions below to rate the description's helpfulness and correctness. Make sure to answer all of the questions. If you can't see the image or caption, press "Image/Caption not visible".

HIT Images Remaining: 10

**Image Caption A:** A Japan Airlines plane taking off from an airport, possibly in Japan, with mountains in the background.
**Image Caption B:** A plane taking off from a runway with mountains in the background.

**Helpfulness:** Which caption provide more useful information for a person who cannot see the image (and who is trying to understand what is in the image)?

Caption A     Caption B     Both are the same

I can't tell

**Correctness:** Which caption is more factually accurate? The caption is...

Caption A     Caption B     Both are the same

I can't tell

Image/Caption Not Visible                    Submit

Figure A12: Description rating tool (head-to-head).