# OpenReview forum: "IC3: Image Captioning by Committee Consensus"
_EMNLP/2023/Conference — EMNLP 2023 Main_

### Official Review · Reviewer_WL9Q · 2023-08-05

**Soundness:** 4

**Excitement:**

4: Strong: This paper deepens the understanding of some phenomenon or lowers the barriers to an existing research direction.

**Paper Topic And Main Contributions:**

This paper presents a relatively simple idea. Generate captions in the traditional way for a given image. Then fold those captions together into a single coherent and detailed caption through the use of what is essentially an LLM summarisation algorithm. While this is a simple idea, it is a good idea and the authors present a nice overview of the approach and detailed human evaluation of the outputs achieved.

**Reasons To Accept:**

While a simplistic approach overall (wisdom of the crowd), the detailed presentation of the model will be beneficial to the controllable image captioning community as a whole. While some papers have looked at slightly similar approaches (as ackknowledged by the authors), this model brings togeter a number of key issues in a fluid way. The human evaluation is also positive.

**Reasons To Reject:**

Some might argue that the approach lacks overall novelty, but I feel that this is outweighed by the quality of some of the results.

**Reproducibility:**

3: Could reproduce the results with some difficulty. The settings of parameters are underspecified or subjectively determined; the training/evaluation data are not widely available.

**Reviewer Confidence:**

3: Pretty sure, but there's a chance I missed something. Although I have a good feel for this area in general, I did not carefully check the paper's details, e.g., the math, experimental design, or novelty.

---

> ### Author Rebuttal · Authors · 2023-08-29
>
> We appreciate the reviewer’s comments, and that the reviewer highlights our simple, fluid and exciting approach to generating image captions, and the human evaluation experiments that we perform to demonstrate their value.
>
> While some may argue that the approach lacks overall novelty, we strongly believe that the technical contribution is significant. We would like to highlight that as far as we are aware:
> - No existing work demonstrates a pipeline for generating a single caption by integrating distributionally-faithful candidate captions. IC3 does this without changing model architecture or retraining by leveraging summarization to produce a single omnibus caption capturing the full distribution of information.
> - No existing work for paragraph captioning or image captioning uses summarization of distributionally-faithful caption samples.
> - No existing work demonstrates in human experiments that long-form captions encoding this distribution are preferable to single reference captions.

---

### Official Review · Reviewer_5Y4s · 2023-08-07

**Soundness:** 3

**Excitement:**

4: Strong: This paper deepens the understanding of some phenomenon or lowers the barriers to an existing research direction.

**Paper Topic And Main Contributions:**

This paper introduces a simple, yet novel method - “Image Captioning by Committee Consensus” which is designed to generate a single caption that captures details from multiple viewpoints by sampling from the learned semantic space of a base captioning model, and also leveraging a LLM to synthesize these samples into a single comprehensive caption.

The key contributions of the paper are –

1. An approach to leverage pretrained image captioning models, and LLM’s to generate semantically complete image captions from a collection of “single-viewpoint” captions.

2. Human evaluation of the method demonstrates that captions generated by IC3 are rated higher in helpfulness (without image context) and correctness.

3. Several automated measures also show that captions generated by IC3 contain significantly more semantic information than baseline captions.


**Questions For The Authors:**

Q1. What is the size of the dataset used for human evaluation?

Q2. How many subjects participated in the human evaluation?


**Reasons To Accept:**

Following are the strengths of the paper -

1. The paper is well written and easy to follow.

2. The approach is very well thought and backed by a detailed human evaluation study which validates it.

3. The automatic evaluation results also support that the IC3 approach significantly outperform SOTA captions.


**Reasons To Reject:**

Following are the weaknesses of the paper -

1. Since the approach uses LLM's, hence it will also suffer from its weakness like hallucinations.

2. The model isn't controllable and fails to describe the background or contextual information.

3. Since the model isn't controllable, it might not generalize to the real world scenarios.


**Reproducibility:**

5: Could easily reproduce the results.

**Reviewer Confidence:**

3: Pretty sure, but there's a chance I missed something. Although I have a good feel for this area in general, I did not carefully check the paper's details, e.g., the math, experimental design, or novelty.

---

> ### Author Rebuttal · Authors · 2023-08-29
>
> We appreciate the reviewer’s comments, and thorough understanding of the paper and the appendix, and that the reviewer highlights our well thought out approach and detailed evaluations. We'd like to take this opportunity to respond to some of the weaknesses and questions discussed by the reviewer:
>
> **LLM Controllability/Hallucination:** While we thoroughly discuss the limitations of our implementation (Sec 6) in terms of hallucination in appendix H.1 and controllability in Appendix H.2, it’s important to distinguish controllability of our method vs the current implementation of it with specific languages models. Both our caption-generating VLM and our summarizing LLM admit prompts which we optimized. So arbitrary controls/direction can be passed to those models. The fact that the models do not always follow such instructions, and regularly hallucinate is a well-known challenge of current LLMs and is the subject of a great amount of research on instruction-tuning, RLHF, and other methods. As hallucination and instruction-following performance of the base models improves, we should inherit similar improvements in IC3.
>
> **Human Evaluation Questions:** While the setup varies slightly between experiments, in general we use the 5000 samples from the MS-COCO karpathy test set (or the 200 sample “hard” subset of that constructed according to our discussion in appendix C.2). The number of subjects is different for each experiment (given by $n$ values in Section 4.1), but ranges from 41 to 121 subjects.

---

### Official Review · Reviewer_k3RY · 2023-08-10

**Soundness:** 3

**Excitement:**

3: Ambivalent: It has merits (e.g., it reports state-of-the-art results, the idea is nice), but there are key weaknesses (e.g., it describes incremental work), and it can significantly benefit from another round of revision. However, I won't object to accepting it if my co-reviewers champion it.

**Missing References:**

The many aspects of this article are very substantial, but there are still some shortcomings in the experimental setup and structural layout. I personally believe that these works will also be helpful to the author, which can refer to the design of these articles in terms of experiments and chapter layout:

[1] Adaptive path selection for dynamic image captioning
[2] Dual global enhanced transformer for image captioning

**Paper Topic And Main Contributions:**

For most "single-viewpoint" captions methods, this article proposes a relatively simple, committee consensus image captioning method, which performs well in some artificial experiments. This manuscript has a lot of content and has designed many experiments. However, this article lacks experiments on core indicators, and it is also recommended to supplement more theoretical analysis.

**Questions For The Authors:**

Suggestion:
1. Adjust the structure of the article. In terms of content, it is very substantial, and the author has analyzed it from multiple perspectives but hopes to highlight the key points.
2. Highlight theoretical highlights. Conduct more theoretical analysis and improve Figure 2 (or add a framework diagram to introduce the model).
3. Supplementary experiments. Add experiments on some cutting-edge indicators in the current image captioning field, as mentioned earlier. Unlike many articles, some commonly used indicator experiments are in the appendix, and it is recommended to provide corresponding supplementary introductions, which improve the readability of this article.

**Reasons To Accept:**

1. Although the author didn’t compare the core indicators in the experiment, they also conducted many experiments to prove their performance from other perspectives.
2. Compared to other single-viewpoint methods, the author's committee consensus method, although relatively simple, has good results.

**Reasons To Reject:**

1. I personally believe that the entire article seems to have been improved and summarized from practical applications. There is relatively little theoretical reasoning in the article, which does not reflect the innovation in the theoretical aspect. And the content of Figure 2 is not concise and clear, it is recommended to supplement the Framework diagram to introduce the model.
2. The biggest problem with the experiments is that although the model proposes a method based on a large model, general indicator analyses such as BLEU-1, BLEU-4, METR, and CIDEr are not shown in the experimental section for Image Captioning tasks. Moreover, no BLIP has online test scores based on the MSCOCO dataset, which are not mentioned in the article. At the same time, there is also a lack of performance comparison with mainstream Image Captioning models in recent years, and the baseline performance of BLIP is not the best in Image Captioning. Without these key experiments and indicators, it is impossible to show the innovation and progressiveness of the methods in the field of Image Captioning.
3. Through Rebuttal, the author provided some supplementary explanations for Weakness 2 (experimental design), but objectively speaking, the readability of this article is somewhat lacking, and some essential details are not handled carefully enough.

**Reproducibility:**

3: Could reproduce the results with some difficulty. The settings of parameters are underspecified or subjectively determined; the training/evaluation data are not widely available.

**Reviewer Confidence:**

4: Quite sure. I tried to check the important points carefully. It's unlikely, though conceivable, that I missed something that should affect my ratings.

---

> ### Author Rebuttal · Authors · 2023-08-29
>
> We appreciate the reviewer’s detailed comments and that the reviewer highlights our simple and performant approach and detailed evaluations. We'd like to take this opportunity to respond to some of the weaknesses and questions discussed by the reviewer:
>
> **Lack of Comparison on Common Indicators (BLEU/METEOR/CIDEr/ROUGE)/Mainstream Image Captioning Models:** As we discuss in L572-L577 of the paper, and in detail in Appendix C.3, traditional “n-gram” based measures such as BLEU(1/4), METEOR, CIDEr, and ROUGE when applied using standard references caption sets are biased towards candidate captions which are impoverished, favoring only “safe” descriptions which are shared among many references. When evaluated using more task- or human-centered metrics, IC3 captions perform better. We argue that this highlights a weakness in those metrics. We nevertheless included those metrics in appendix C3.
>
> In terms of models, the IC3 meta-approach should be transferable to any distributionally-faithful image captioning method (which we validate with TRM measures in Section 3.2). According to the official MS-COCO online benchmark (https://competitions.codalab.org/competitions/3221#results), OFA (discussed in the paper) is ranked #2 only behind GIT which is not publicly available. We also will provide additional results on BLIP-2 (a recent model outperforming BLIP) on the Karpathy Test Set of MS-COCO in the camera ready paper:
>
> _Human Study (N=108, p=0.003 (helpfulness), p=0.898 (correctness)):_
> $$
> \begin{array}{lcc}
> \textbf{Method} & \textbf{Helpfulness} & \textbf{Correctness} \\\\
> \text{BLIP-2 + IC3} & 51.97\\% & 44.10\\% \\\\
> \text{BLIP-2} & 37.49\\% & 42.67\\% \\\\
> \text{Tie} & 9.78\\% & 11.6\\% \\\\
> \end{array}
> $$
>
> _Automated Measures:_
> $$
> \begin{array}{lcccccccc}
> \textbf{Method} & \textbf{Noun Recall} & \textbf{Verb Recall} & \textbf{Fuzzy Noun} & \textbf{Fuzzy Verb} & \textbf{MRR} & \textbf{R@1} & \textbf{R@5} & \textbf{R@10} \\\\
> \text{BLIP-2 + IC3} & 0.364 & 0.229 & 0.667 & 0.529 & 0.746 & 0.652 & 0.863 & 0.921 \\\\
> \text{BLIP-2} & 0.277 & 0.185 & 0.582 & 0.442 & 0.589 & 0.473 & 0.725 & 0.811 \\\\
> \end{array}
> $$
>
> In the case where real human captions are used instead of captioning model output, we still demonstrate in both human studies and automated measures that IC3-summarization outperforms baseline references. In addition, we provide a further head-to-head investigation of references vs. references + IC3 on the Karpathy Test Set of MS-COCO (N=108, p=1.732e-5 (helpfulness), p=0.0019 (correctness)):
>
> $$
> \begin{array}{lcc}
> \textbf{Method} & \textbf{Helpfulness} & \textbf{Correctness} \\\\
> \text{Ref + IC3} & 55.79\\% & 48.8\\% \\\\
> \text{Ref} & 36.65\\% & 36.27\\% \\\\
> \text{Tie} & 7.46\\% & 13.97\\% \\\\
> \end{array}
> $$
>
> **Lack of Theoretical Reasoning:** Our paper uses novel theoretical tools introduced in prior work, namely sparse distributional metrics, and applies them to the practical problem of caption generation. Since the implementation and performance of those metrics was described in another paper, we did not repeat it here. However, those metrics have not previously been applied to captioning, and that requires a robust summarization method, which  is the subject of the present paper. We want to emphasize that there are two key lessons from this work (i) that there is more useful information in _collections_ of human captions than in a single caption (the “C” for Committee in the paper title), and that (ii) optimizing a distributionally-faithful baseline model generating captions at temperature near 1.0 reproduces this human diversity, allowing an LLM to integrate this information into a better single caption. Furthermore, as mentioned previously, we pioneer the use of distributional metrics (TRM-based in section 3.2) to allow optimization of the first-stage captioning model for the IC3 application.
>
> In summary, as far as we are aware:
> - No existing work demonstrates a pipeline for generating a single caption by integrating distributionally-faithful candidate captions. IC3 does this without changing model architecture or retraining by leveraging summarization to produce a single omnibus caption capturing the full distribution of information.
> - No existing work for paragraph captioning or image captioning uses summarization of distributionally-faithful caption samples.
> - No existing work demonstrates in human experiments that long-form captions encoding this distribution are preferable to single reference captions.
>
> We will clarify these details in the camera ready version of the paper by expanding our discussion in Sections 1, 2 and 3 to include a more thorough description of the motivation/approach. If space allows, we will add an additional figure to demonstrate the key technical motivation of our approach  to complement Figure 2 (the method figure).

---

### Official Review · Reviewer_9GXH · 2023-08-11

**Soundness:** 3

**Excitement:**

3: Ambivalent: It has merits (e.g., it reports state-of-the-art results, the idea is nice), but there are key weaknesses (e.g., it describes incremental work), and it can significantly benefit from another round of revision. However, I won't object to accepting it if my co-reviewers champion it.

**Paper Topic And Main Contributions:**

This paper uses a prompt engineering approach to generate a summarized long caption from short captions generated by human or image captioning models (BLIP and OFA) using OpenAI GPT APIs. The authors showed that the summarized captions are better than the ones from human, BLIP, and OFA on both human evaluation (AMT) and automated evaluation.

**Questions For The Authors:**

What is the improvement of IC3 on top of the recent image captioning model, such as BLIP-2?

**Reasons To Accept:**

This paper proposes a simple but exciting approach to generating fine-grained image captions using OpenAI GPT APIs. They conducted detailed experiments to show the values of generated captions on both human and automated evaluations. The description rating tool is potentially helpful for similar tasks.

**Reasons To Reject:**

The approach is prompt engineering and lacks comparison to LLM-based image captioning models such as BLIP-2.

**Reproducibility:**

5: Could easily reproduce the results.

**Reviewer Confidence:**

3: Pretty sure, but there's a chance I missed something. Although I have a good feel for this area in general, I did not carefully check the paper's details, e.g., the math, experimental design, or novelty.

---

> ### Author Rebuttal · Authors · 2023-08-29
>
> We appreciate the reviewer’s comments, and that the reviewer highlights our simple and exciting approach to generating fine-grained image captions, and the detailed experiments that we perform to demonstrate their value. We'd like to take this opportunity to respond to some of the weaknesses and questions discussed by the reviewer:
>
> **Performance on BLIP-2/SOTA Models:** We provide the results on BLIP-2 (both the automated measures and an additional human head to head evaluation) on the Karpathy Test Set of MS-COCO below, and will include them in the camera ready version of the paper. The results for BLIP-2 are in line with BLIP-1 and OFA; humans prefer IC3 augmented captions in terms of helpfulness, and find them as correct as BLIP-2 captions (no regression in correctness), and IC3 results in significantly improved recall over baselines.
>
> _Human Evaluation (N=108, p=0.003 (helpfulness), p=0.898 (correctness)):_
> \begin{array}{lcc}
> \textbf{Method} & \textbf{Helpfulness} & \textbf{Correctness} \\\\
> \text{BLIP-2 + IC3} & 51.97\\% & 44.10\\% \\\\
> \text{BLIP-2} & 37.49\\% & 42.67\\% \\\\
> \text{Tie} & 9.78\\% & 11.6\\% \\\\
> \end{array}
>
> _Automated Measures:_
> \begin{array}{lcccccccc}
> \textbf{Method} & \textbf{Noun Recall} & \textbf{Verb Recall} & \textbf{Fuzzy Noun} & \textbf{Fuzzy Verb} & \textbf{MRR} & \textbf{R@1} & \textbf{R@5} & \textbf{R@10} \\\\
> \text{BLIP-2 + IC3} & 0.364 & 0.229 & 0.667 & 0.529 & 0.746 & 0.652 & 0.863 & 0.921 \\\\
> \text{BLIP-2} & 0.277 & 0.185 & 0.582 & 0.442 & 0.589 & 0.473 & 0.725 & 0.811 \\\\
> \end{array}
>
> In the case where real human captions are used instead of captioning model output, we still demonstrate in both human studies and automated measures that IC3-summarization outperforms baseline references. In addition, we provide a further head-to-head investigation of references vs. references + IC3 on the Karpathy Test Set of MS-COCO  (N=108, p=1.732e-5 (helpfulness), p=0.0019 (correctness)):
>
> \begin{array}{lcc}
> \textbf{Method} & \textbf{Helpfulness} & \textbf{Correctness} \\\\
> \text{Ref + IC3} & 55.79\\% & 48.8\\% \\\\
> \text{Ref} & 36.65\\% & 36.27\\% \\\\
> \text{Tie} & 7.46\\% & 13.97\\% \\\\
> \end{array}
>
> **Method is primarily prompt engineering:** We want to emphasize that there are two key lessons from this work (i) that there is more useful information in _collections_ of human captions than in a single caption (the “C” for Committee in the paper title), and that (ii) optimizing a distributionally-faithful baseline model generating captions at temperature near 1.0 reproduces this human diversity, allowing an LLM to integrate this information into a better single caption. Furthermore, beyond the summarization stage, we pioneer the use of distributional metrics (TRM-based in section 3.2) to allow optimization of the first-stage captioning model for the IC3 application.
>
> In summary, as far as we are aware:
> - No existing work demonstrates a pipeline for generating a single caption by integrating distributionally-faithful candidate captions. IC3 does this without changing model architecture or retraining by leveraging summarization to produce a single omnibus caption capturing the full distribution of information.
> - No existing work for paragraph captioning or image captioning uses summarization of distributionally-faithful caption samples.
> - No existing work demonstrates in human experiments that long-form captions encoding this distribution are preferable to single reference captions.
>
> We will clarify these details in the camera ready version of the paper by expanding our discussion in Sections 1, 2 and 3 to include a more thorough description of the motivation/approach. If space allows, we will add an additional figure to demonstrate the key technical motivation of our approach  to complement Figure 2 (the method figure).

---

### Meta-Review · Area_Chair_pge1 · 2023-09-16

**Recommendation:** 5

**Metareview:**

This paper proposes a simple method to generate a single coherent and detailed image caption through the use of LLM to summarize outputs from multiple generated short image captions. The authors have done a nice job of rebuttal.

In general, 3 reviewers are positive about the paper, and one reviewer is slightly negative about the paper. Reviewers found that (1) the proposed method is simple and exciting; (2) experiments are comprehensive with detailed human evaluation study; (3) the paper is well written. One reviewer shared the concern that the standard evaluation metrics like BLEU/CIDEr scores are not reported and there is a lack of comparison with most SoTA image captioning models. The AC thinks that this is not a major concern, since the proposed method aims to provide a detailed comprehensive caption, rather than a short caption that aims to chase SoTA CIDEr score. Overall, given the reviews, the AC is positive about the paper.

---

### Decision · Program_Chairs · 2023-10-07

**Decision:**

Accept-Main

**Comment:**

This paper proposes a simple method to generate a single coherent and detailed image caption through the use of LLM to summarize outputs from multiple generated short image captions. The authors have done a nice job of rebuttal.

In general, 3 reviewers are positive about the paper, and one reviewer is slightly negative about the paper. Reviewers found that (1) the proposed method is simple and exciting; (2) experiments are comprehensive with detailed human evaluation study; (3) the paper is well written. One reviewer shared the concern that the standard evaluation metrics like BLEU/CIDEr scores are not reported and there is a lack of comparison with most SoTA image captioning models. The AC thinks that this is not a major concern, since the proposed method aims to provide a detailed comprehensive caption, rather than a short caption that aims to chase SoTA CIDEr score. Overall, given the reviews, the AC is positive about the paper.